# FlowMAP: Flow Matching for Generalizable Agent Planning

**Jiarun Fu** [1]  **Lizhong Ding** [1]  **Ye Yuan** [2]  **Qiuning Wei** [1]  **Zhaohuan Linghu** [1]  **Yurong Cheng** [1]  **Changsheng Li** [1]
**Tianlong Gu** [3]  **Liang Chang** [4]  **Guoren Wang** [1]

{jrfu, yuan-ye, weiqiuning, zhaohuan-linghu, yrcheng, lcs}@bit.edu.cn, lizhong.ding@outlook.com,
gutianlong@jnu.edu.cn, changl@guet.edu.cn, wanggrbit@126.com

## Abstract

Agent planning faces dynamic heterogene­ity—nonstationary observations, dynamics, and objectives with sparse, delayed rewards—which dominant methods largely ignore, leading to poor generalization under environment shifts. We propose *Flow-Matching for Agent Planning* (FlowMAP), which formulates planning as a continuous-time flow-matching problem by learn­ing a planning-time velocity field that transports an initial meta-state distribution toward a task-conditioned target. FlowMAP introduces Value-Transport Flow Matching to provide a distribution-level planning objective that steers transport to­ward high-value regions in the meta-state dis­tribution, mitigating error accumulation under environmental shifts. To enforce alignment be­tween meta-state distribution transport and action–environment interaction, FlowMAP further pro­poses Flow–Policy Co-Training, which jointly optimizes the planning flow and policy so that the flow transport directly regularizes the policy-induced meta-distribution dynamics. Across di­verse agent planning benchmarks, FlowMAP con­sistently outperforms strong baselines, yielding improvements in planning generalization.

## 1. Introduction

Agent planning requires policies that generate long-horizon action sequences under partial observability while remain­ing effective as the environment changes (Guo et al., 2024; Ziliotto et al., 2025). In practice, agents may face dynamic heterogeneity (e.g., varying layouts, changing object place­ments and affordances, different game rules, or stochastic transitions) (Matthews et al., 2024; Bonnet et al., 2023; Nikulin et al., 2025), as well as observation variations (e.g., sensor noise, lighting, occlusions, or missing fields) (Chung et al., 2024; Mon-Williams et al., 2025). Such *dynamic het­erogeneity* induces distribution shift between training and de­ployment, while the sparse rewards pervasive in open-world settings make training signals delayed and weak (Hafner et al., 2025; Hamadanian et al., 2025); early exploratory choices can have irreversible downstream consequences, and even modest shifts can compound over time (Sims et al., 2024; Li et al., 2026; Guo et al., 2026), ultimately prevent­ing planning policies from generalizing to environmental changes and making generalizable planning a core bottle­neck for embodied agents, multi-agent coordination, and tool-using workflow agents (Zhou et al., 2024b; Patel et al., 2024). Therefore, achieving environment-generalizable agent planning remains an urgent research objective.

Recent work toward environment-generalizable planning largely follows two complementary routes—(i) step-by-step action optimization via *model-free reinforcement learning* and (ii) *planning with learned world models*—with progres­sively stronger designs emerging within each (Sutton et al., 1998; Mnih et al., 2015; Hafner et al., 2025; Fu et al., 2026). On the model-free reinforcement learning side, methods strengthen exploration to better cope with partial observabil­ity and long-horizon planning: RESeL stabilizes recurrent off-policy learning by decoupling optimization dynamics for the context encoder and policy head (Luo et al., 2024); Memory Traces compress history into trace-like summaries to make long contexts trainable (Eberhard et al., 2025); and EME defines intrinsic bonuses using principled state-discrepancy metrics to improve exploration (Wang et al., 2024). However, these methods remain brittle under dynam­ics shifts, where early exploratory choices can irreversibly steer trajectories and errors compound over time (Zheng et al., 2024). World-model approaches move beyond purely step-wise updates by learning predictive latent dynamics

---

[1]School of Computer Science and Technology, Beijing Institute of Technology, Beijing, China [2]School of Artificial Intelligence, Beijing Institute of Technology, Beijing, China [3]College of Infor­mation Science and Technology/ College of Cyber Security, Jinan University, Guangzhou, China [4]Guangxi Key Lab of Trusted Soft­ware, Guilin University of Electronic Technology, Guilin, China. Correspondence to: Lizhong Ding <lizhong.ding@outlook.com>.

*Proceedings of the 43rd International Conference on Machine Learning*, Seoul, South Korea. PMLR 306, 2026. Copyright 2026 by the author(s).

and optimizing policies through imagination or planning in the learned simulator, as exemplified by DreamerV3 (Hafner et al., 2025), RoboDreamer (Zhou et al., 2024a), and Vista (Gao et al., 2024). However, under dynamic heterogeneity, the learned simulator is inherently *regime-specific*: being calibrated to the training distribution, even mild environment shifts and state representation mismatch can compound with rollout depth, causing imagined trajectories to drift and ultimately undermining environment-generalizable planning, as shown in Figure 1 (Ramasubramanian et al., 2026; Wang et al., 2025; Lin et al., 2025; McAllister et al., 2026). We summarize the challenge of environment-generalizable agent planning:

> The lack of agent-planning methods that explicitly model and control *dynamic heterogeneity* to enable environment-generalizable planning.

This paper takes a different viewpoint: under *dynamic heterogeneity*, the key object to control is not a single predicted trajectory, but the *meta-distribution* over the agent's internal state representation. The meta-state inferred from interaction history implicitly encodes hypotheses about latent environment factors; consequently, a policy induces a meta-state occupancy distribution. Generalization under shift then hinges on whether this distribution can be reliably transported toward high-value regions despite changing dynamics and observations.

Flow matching offers a sample-based mechanism for distributional control, learning a continuous-time velocity field whose induced flow transports one distribution to another and specifies global probability-mass movement in representation space (Lipman et al., 2023; Dao et al., 2023). So we propose Flow Matching for generalizable Agent Planning (**FlowMAP**), which formulates planning as continuous-time flow matching and directly learns a planning-time velocity field to transport an agent's initial meta-state distribution toward a task-conditioned target meta-distribution. To provide a scalable, distribution-level planning signal under dynamic heterogeneity, FlowMAP introduces Value-Transport Flow Matching (VTFM), which constructs a value-shaped target distribution and an efficient coupling that pairs meta-states with high-reward regions, supplying a generalizable training signal even when rewards are sparse and shifts compound over long horizons. To enforce alignment between meta-state distribution transport and action-environment interaction, we further propose Flow-Policy Co-Training (FPCT), which jointly optimizes the planning flow and the policy with shared representations so that the learned transport induces actions that reliably drive the meta-state distribution toward the target. Across diverse open-world benchmarks, FlowMAP outperforms strong world-model and reinforcement-learning baselines, demonstrating substantial gains in environment-generalizable planning.

Our contributions can be summarized as follows:

1. We propose Flow-Matching for Agent Planning (FlowMAP), which formulates agent planning under dynamic heterogeneity as continuous-time flow matching over meta-state distributions by learning a planning-time velocity field that transports an initial meta-state distribution toward a task-conditioned target, improving generalization under environment shifts.

2. We introduce Value-Transport Flow Matching to provide a distribution-level planning objective under dynamic heterogeneity, steering meta-state distribution transport toward high-value regions to mitigate error accumulation induced by environment shifts.

3. To align meta-state distribution transport with action–environment interaction, we propose Flow–Policy Co-Training, jointly optimizing the planning flow and policy so that the flow transport directly regularizes policy-induced meta-distribution dynamics, yielding more reliable planning generalization.

## 2. Preliminaries

### 2.1. POMDPs and Belief MDPs

The interaction dynamics between the agent and the environment for each task $l \in L$ is usually formulated as a Partially Observable Markov Decision Process (POMDP):

$$\mathcal{M}^l = (\mathcal{X}, A, \mathcal{O}, F_l, O_l, R_l, \gamma, \mu_0^l),$$

where $\mathcal{X}$ is the latent environment state space ($x_t \in \mathcal{X}$), $A$ is the action space ($a_t \in A$), $\mathcal{O}$ is the observation space ($o_t \in \mathcal{O}$), $F_l(x_{t+1} \mid x_t, a_t)$ is the task-conditioned transition function, $O_l(o_t \mid x_t, a_{t-1})$ is the observation function, $R_l(x_t, a_t)$ is the reward, $\gamma \in (0, 1)$ is the discount, and $\mu_0^l$ is the initial state distribution.

A trajectory $(x_0, o_0, a_0, x_1, o_1, a_1, \dots)$ is generated by

$$x_0 \sim \mu_0^l, \quad o_t \sim O_l(\cdot \mid x_t, a_{t-1}),$$
$$a_t \sim \pi_\theta(\cdot \mid h_t, l), \quad x_{t+1} \sim F_l(\cdot \mid x_t, a_t),$$

where $h_t = (o_0, a_0, \dots, o_{t-1}, a_{t-1}, o_t)$ is the history and $\pi_\theta$ is a history-dependent policy. The objective is to maximize the expected discounted return

$$J(\pi_\theta \mid l) = \mathbb{E}_{\pi_\theta}\left[\sum_{t=0}^{\infty} \gamma^t R_l(x_t, a_t)\right].$$

### 2.2. Flow Matching

Flow matching provides a sample-based approach to learn distributional transport in continuous time. Let $X \subset \mathbb{R}^d$ and

*Figure 1.* Conceptual comparison of agent-planning paradigms under environmental shifts. (1) Model-free RL relies on step-wise policy/value updates from an implicit history state, but it does not explicitly handle dynamic heterogeneity or observation variations, so behavior can break under shifts. (2) World-model planning imagines action-conditioned rollouts in a learned latent space, which helps with observation variations, yet it typically assumes fixed latent dynamics and can drift when regimes change. **(3) FlowMAP (Ours)** addresses both by shaping how the agent's meta-state distribution evolves under changing observations and dynamics, letting it track and steer its internal hypotheses and yield more generalizable planning in heterogeneous environments.

let $\{\rho_\tau\}_{\tau \in [0,1]}$ interpolate between $\rho_0$ and a desired target $\rho_\star$. A time-dependent velocity field $v : X \times [0,1] \to \mathbb{R}^d$ defines a probability flow through the continuity equation

$$\partial_\tau \rho_\tau(x) \; + \; \nabla_x \cdot \big(\rho_\tau(x)\, v(x,\tau)\big) \; = \; 0, \qquad (1)$$

where $\rho_0$ is given and $\rho_1 \approx \rho_\star$. The corresponding flow map $\Phi_\tau$ solves the ODE:

$$\frac{d}{d\tau}\, \Phi_\tau(x) \; = \; v(\Phi_\tau(x), \tau), \qquad \Phi_0(x) = x,$$

and transports $\rho_0$ by pushforward: $\rho_\tau = (\Phi_\tau)_\sharp \rho_0$.

In flow matching, $v$ is parameterized by $v_\theta$ and trained from samples. Given a coupling $\eta \in \Pi(\rho_0, \rho_\star)$ and pairs $(x_0, x_1) \sim \eta$, define an interpolant

$$x_\tau \; = \; x_0 + \alpha(\tau)(x_1 - x_0), \qquad \alpha(0) = 0, \; \alpha(1) = 1,$$

with reference velocity

$$v^*(x_\tau, \tau \mid x_0, x_1) \; = \; \alpha'(\tau)(x_1 - x_0).$$

Flow matching minimizes

$$\mathcal{L}_{\mathrm{FM}}(\theta) \; = \; \mathbb{E}_{(x_0,x_1) \sim \eta}\, \mathbb{E}_{\tau \sim \mathcal{U}[0,1]} \Big\| v_\theta(x_\tau, \tau) - v^*(x_\tau, \tau \mid x_0, x_1) \Big\|^2, \tag{2}$$

which, under mild conditions, yields a velocity field whose induced flow transports $\rho_0$ toward $\rho_\star$ in the sense of (1).

GFlowNets, FlowRL, and ReinFlow all recast decision making as matching a reward-shaped target distribution over terminal outcomes, reward distributions, or action distributions—using flow-based objectives to improve mode coverage (Bengio et al., 2021; Zhu et al., 2026; Zhang et al., 2025); however, they primarily shape *what* is sampled rather than explicitly controlling *how* probability mass redistributes over the agent's evolving internal state under dynamic heterogeneity, and thus do not directly regulate state-distribution shift induced by environment dynamics (see Appendix A.1). While recent agent-planning work often incorporates LLMs, this paper focuses on reinforcement-learning-based planning under dynamic heterogeneity, with further scope discussion in Appendix A.2.

## 3. FlowMAP

### 3.1. Planning as Continuous-Time Flow Matching

Agent planning tasks exhibit *dynamic heterogeneity*: observations, transition dynamics, and reward signals can vary across scenarios and evolve over time. We index each task $l$ by a latent dynamical factor $\xi_l \in \Xi$ such that

$$\begin{aligned} F_l(\cdot \mid x_t, a_t) &\equiv F(\cdot \mid x_t, a_t, \xi_l), \\ O_l(\cdot \mid x_t, a_{t-1}) &\equiv O(\cdot \mid x_t, a_{t-1}, \xi_l), \\ R_l(x_t, a_t) &\equiv R(x_t, a_t, \xi_l). \end{aligned}$$

In open-world settings, the latent environment factor $\xi_l \in \Xi$ is often difficult to model explicitly: it is typically unob-

served, high-dimensional, and may vary across tasks and over time, while being entangled with the transition and observation mechanisms. With limited supervision, any explicit parametric model of $\xi_l$ is prone to misspecification, which in turn prevents us from directly characterizing environment shifts at the level of $\xi_l$.

Under dynamic heterogeneity and partial observability, generalizable planning requires an internal, updateable summary of interaction history that captures the agent's evolving hypothesis about the latent regime. We therefore take $s_t \in \mathbb{R}^{d_S}$ as a learned meta-state embedding inferred from the observation–action history, serving as a compact approximate sufficient statistic for decision making under shift. Although $s_t$ is not an explicit belief distribution, it can be viewed as implicitly encoding the agent's posterior hypothesis over latent environment factors (e.g., $\xi_l$) through representation learning, thereby representing heterogeneity in a tractable Euclidean space $\mathcal{S} \subseteq \mathbb{R}^{d_S}$. Motivated by this perspective, FlowMAP formulates agent planning as distributional control over meta-state occupancies, and instantiates this control via continuous-time flow matching. For a fixed task $l$ and policy $\pi_\theta$, define the meta-state distribution at environment time $t$ as

$$\rho_t^l(s) \;=\; p(s_t = s \mid \pi_\theta, l), \qquad s \in \mathcal{S}. \qquad (3)$$

The meta-state evolves according to an induced Markov transition function

$$s_{t+1} \sim P_{\text{meta}}(\,\cdot \mid s_t, a_t, l), \qquad a_t \sim \pi_\theta(\,\cdot \mid s_t, l), \quad (4)$$

where $P_{\text{meta}}(\,\cdot \mid s_t, a_t, l)$ denotes the task-conditioned *meta-state transition law* induced jointly by (i) the underlying POMDP dynamics specified by the task-conditioned transition and observation functions, i.e., $F_l(\,\cdot \mid x_t, a_t)$ and $O_l(\,\cdot \mid x_t, a_{t-1})$, and (ii) the agent's internal representation embedded in the policy model $\pi_\theta$, which summarizes the interaction history into the meta-state $s_t$ and then selects actions as $a_t \sim \pi_\theta(\,\cdot \mid s_t, l)$. Throughout, we treat $s_t$ as an approximate sufficient statistic so that the induced meta-state dynamics can be modeled as Markov in $\mathcal{S}$. Intuitively, after executing $a_t$, the environment evolves according to the $\xi_l$-dependent dynamics, new observations are produced, and the agent updates its internal meta-state distributions, resulting in $s_{t+1}$:

$$\rho_{t+1}^l(s') \;=\; \int_\mathcal{S} \rho_t^l(s) \int_A \pi_\theta(a \mid s, l) \, P_{\text{meta}}(s' \mid s, a, l) \, \mathrm{d}a \, \mathrm{d}s, \qquad (5)$$

Eq. (5) characterizes how the meta-distribution $\rho_t^l$ evolves over environment time $t$ once a policy $\pi_\theta$ is executed. Under dynamic heterogeneity, generalizable planning requires actively shaping how probability mass over internal hypotheses redistributes in $\mathcal{S}$, rather than merely tracking its passive

evolution. Motivated by this perspective, FlowMAP *formulates* agent planning as continuous-time flow matching over meta-state distributions, learning an explicit transport that maps an initial meta-distribution $\rho_0^l$ to a task-conditioned target meta-distribution $\rho_\star^l$. To explicitly control probability redistribution in $\mathcal{S}$, we introduce an auxiliary planning time $\tau \in [0,1]$ and model the planning-time evolution of $\rho_\tau^l$ by the continuity equation:

$$\partial_\tau \rho_\tau^l(s) \;+\; \nabla_s \cdot \big( \rho_\tau^l(s) \, v_\theta(s, \tau, l) \big) \;=\; 0, \qquad (6)$$

which enforces conservation of probability mass while transporting $\rho_\tau^l$ along the velocity field $v_\theta$ in $\mathcal{S}$. Here, $\rho_\tau^l(s)$ denotes the meta-state distribution at planning time $\tau$, and $v_\theta(s, \tau, l)$ specifies how probability mass is advected locally around $s$ under task $l$. Concretely, at environment time $t$ we set the initial condition as $\rho_0^l := \rho_t^l$ (the current meta-distribution induced by $\pi_\theta$), and evolve (6) over $\tau \in [0,1]$ so that the terminal distribution satisfies $\rho_1^l \approx \rho_\star^l$, i.e., the transported meta-distribution approaches a task-conditioned target. We parameterize a task-conditioned velocity field

$$v_\theta : \mathcal{S} \times [0,1] \times L \to \mathbb{R}^{d_S}, \qquad (s, \tau, l) \mapsto v_\theta(s, \tau, l), \;\; (7)$$

and model the planning-time evolution according to the continuity equation (6).

Equivalently, the flow map $\Phi_\tau^l : \mathcal{S} \to \mathcal{S}$ solves

$$\frac{d}{d\tau} \Phi_\tau^l(s) \;=\; v_\theta(\Phi_\tau^l(s), \tau, l), \qquad \Phi_0^l(s) = s, \quad (8)$$

and transports $\rho_0^l$ by pushforward: $\rho_\tau^l = (\Phi_\tau^l)_\sharp \rho_0^l$. Thus, FlowMAP formulates agent planning as distributional control over meta-state occupancies by learning a task-conditioned planning-time transport, which subsequently serves as a structured regularizer for policy learning under dynamic heterogeneity and environment shifts.

### 3.2. Value-Transport Flow Matching (VTFM)

The formulation above reduces planning to learning a task-conditioned transport on $\mathcal{S}$, but it leaves one key question open: *what meta-distribution should the transport aim for?* Under dynamic heterogeneity, a naive choice such as matching the next-step meta-distribution induced by the current policy (i.e., merely fitting $\rho_{t+1}^l$) provides no explicit preference over which internal hypotheses and meta-states should be emphasized, and thus cannot reliably steer probability mass toward behaviors that remain effective under environment shifts. To make the transport operational for decision making, the target meta-distribution must encode the task objective—it should assign higher mass to meta-states that are both consistent with the agent's evolving hypotheses and predictive of high long-horizon return.

Motivated by this necessity, we propose *Value-Transport Flow Matching (VTFM)*, which uses value estimates only to

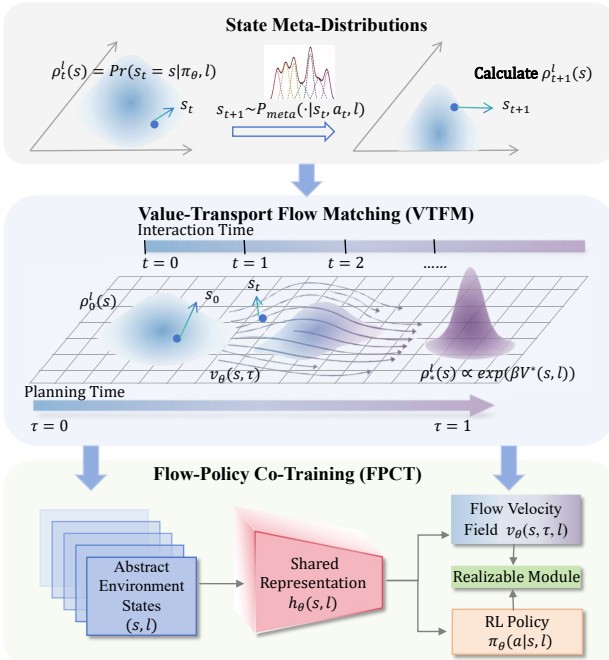

*Figure 2.* **FlowMAP pipeline overview.** FlowMAP models interaction-induced meta-state occupancies as distributions $\rho_t^l$ that evolve over environment time. It then introduces a planning-time flow: Value-Transport Flow Matching (VTFM) learns a task-conditioned velocity field $v_\theta(s, \tau, l)$ to transport the current $\rho_0^l$ toward a value-shaped target $\rho_\star^l$. Finally, Flow–Policy Co-Training (FPCT) couples the flow and the RL policy via a shared representation, aligning the learned transport directions with the meta-state dynamics induced by action–environment transitions.

shape a target distribution over meta-states, rather than to perform greedy or pointwise value-based control.

Let $V^*(s, l)$ be the optimal value functional on meta-states $s \in \mathcal{S}$ under task $l$, defined as

$$V^*(s, l) := \sup_\pi \mathbb{E}\left[\sum_{k=0}^\infty \gamma^k R_l(x_{t+k}, a_{t+k}) \,\middle|\, s_t = s,\ l\right],\tag{9}$$

where $\gamma \in (0, 1)$ is the discount factor, and the conditioning on $s_t$ emphasizes that the value is evaluated with respect to the latent-state belief/hypothesis encoded by the meta-state. We instantiate the target meta-distribution via value shaping and choose a soft-optimal target

$$\rho_\star^l(s) \propto \exp\big(\beta V^*(s, l)\big), \qquad \beta > 0,\tag{10}$$

where $\beta$ controls the sharpness of the target (larger $\beta$ yields a more concentrated target on high-value meta-states). In practice, $V^*(s, l)$ is not accessible and we approximate it with a learned critic $\hat{V}_\psi(s, l)$ from the RL backbone. As $\rho_\star^l$ is value-shaped, the learned transport provides an explicit distribution-level control signal that stabilizes planning under dynamic heterogeneity, without requiring a well-calibrated critic.

To learn the planning flow without solving optimal-control PDEs, we apply conditional flow matching. Let $\Pi(\rho_0^l, \rho_\star^l)$ denote the set of couplings (joint distributions) whose marginals are $\rho_0^l$ and $\rho_\star^l$, and let $\eta^l \in \Pi(\rho_0^l, \rho_\star^l)$ be a coupling between the initial and target meta-distributions.[1] To make the pairing interpretable and stable under heterogeneity, we adopt a cost-regularized transport principle:

$$\eta^l = \arg\min_{\eta \in \Pi(\rho_0^l, \rho_\star^l)} \mathbb{E}_{(s^{(0)}, s^{(1)}) \sim \eta}\big[\|s^{(0)} - s^{(1)}\|_2^2\big] - \varepsilon\,\mathcal{H}(\eta),\tag{11}$$

where $\varepsilon > 0$, $\|\cdot\|_2$ is the Euclidean norm, and $\mathcal{H}(\eta)$ is the entropy of the coupling. Here $(s^{(0)}, s^{(1)})$ are *planning-time endpoint samples* (source, target) rather than consecutive environment-time states.

Given $(s^{(0)}, s^{(1)}) \sim \eta^l$, define an interpolant

$$s_\tau = s^{(0)} + \alpha(\tau)\,(s^{(1)} - s^{(0)}), \tau \in [0, 1], \alpha(0) = 0,\ \alpha(1) = 1,\tag{12}$$

where $\alpha : [0, 1] \to [0, 1]$ is a smooth interpolation schedule (e.g., $\alpha(\tau) = \tau$). The corresponding *reference velocity* induced by this interpolant is

$$v^{\mathrm{ref}}(s_\tau, \tau \mid s^{(0)}, s^{(1)}) = \alpha'(\tau)\,(s^{(1)} - s^{(0)}).$$

*Note that $V^*(\cdot, l)$ is a scalar value functional used to define the target $\rho_\star^l$, whereas $v^{\mathrm{ref}}$ is a vector-valued reference velocity used for flow matching; they are conceptually distinct.*

To operationalize the planning-time formulation in (6), we learn a task-conditioned velocity field $v_\theta(s, \tau, l)$ whose induced flow transports $\rho_0^l$ toward the value-shaped target $\rho_\star^l$. Directly solving the continuity equation is intractable in high-dimensional $\mathcal{S}$. Instead, we adopt a task-conditioned flow-matching objective based on paired endpoint samples and a simple interpolant, yielding a scalable supervised regression target for learning distributional transport in high-dimensional $\mathcal{S}$. We minimize

$$\mathcal{L}_{\mathrm{FM}}(\theta) = \mathbb{E}_{l \sim p(L)}\,\mathbb{E}_{(s^{(0)}, s^{(1)}) \sim \eta^l}\,\mathbb{E}_{\tau \sim \mathcal{U}[0,1]}\Big\|v_\theta(s_\tau, \tau, l) - v^{\mathrm{ref}}(s_\tau, \tau \mid s^{(0)}, s^{(1)})\Big\|^2,\tag{13}$$

where $p(L)$ is the task sampling distribution, and $\mathcal{U}[0, 1]$ is the uniform distribution over planning time.

Minimizing (13) fits $v_\theta$ to match the local transport directions specified by the coupling $\eta^l$ and interpolant $\alpha(\tau)$, so that integrating the learned velocity field yields a global redistribution of probability mass consistent with the continuity equation (6). Crucially, because $\rho_\star^l$ is shaped by $V^*(\cdot, l)$, the resulting transport provides an explicit distribution-level control signal, which stabilizes planning under dynamic heterogeneity without requiring a well-calibrated critic.

---

[1] In practice, at environment time $t$ we typically set $\rho_0^l := \rho_t^l$ as the current meta-distribution induced by $\pi_\theta$.

## 3.3. Flow–Policy Co-Training (FPCT)

VTFM learns a *planning-time* transport that reshapes the meta-distribution $\rho_0^l$ toward the value-shaped target $\rho_\star^l$ via the velocity field $v_\theta(s, \tau, l)$. However, interaction with the environment still occurs in *environment time* $t$ through feasible transitions induced by actions. Thus, to turn distributional transport into effective control, we require that policy-induced meta-state evolution remains compatible with the learned transport directions. We therefore propose Flow–Policy Co-Training (FPCT), which jointly optimizes the planning flow and control policy with shared representations. We introduce a shared representation network

$$h_\theta : \mathcal{S} \times L \to \mathbb{R}^m, \tag{14}$$

and define a flow head and a policy head as

$$v_\theta(s, \tau, l) = g_v\big(h_\theta(s, l), \tau\big) \in \mathbb{R}^{d_S}, \tag{15}$$

$$\pi_\theta(a \mid s, l) \propto \exp\Big(g_\pi\big(h_\theta(s, l)\big)^\top \psi(a)\Big), \tag{16}$$

where $g_v$ and $g_\pi$ are learnable heads, $m$ is the representation dimension, and $d_S$ is the meta-state dimension ($s \in \mathbb{R}^{d_S}$). Here $\psi(a)$ denotes an action embedding (e.g., a fixed one-hot embedding for discrete $A$, or a learned feature map for continuous actions). With this parameterization, gradients from the distributional planning loss (13) update the shared representation $h_\theta$, thereby shaping the geometry of $\mathcal{S}$ in a way that is aligned with value-transport.

FlowMAP provides a distribution-level transport signal in planning time $\tau$, while environment interaction is constrained by feasible transitions. To connect transport directions with realizability, we introduce a dynamics-consistency regularizer using a learned one-step representation predictor from the backbone. Concretely, let

$$\hat{s}_{t+1} = f_\phi^{\text{pred}}(s_t, a_t, l) \tag{17}$$

be the predicted next meta-state under the replayed action $a_t$. We define the realized one-step drift in $\mathcal{S}$ as $\Delta s_t := \hat{s}_{t+1} - s_t$. Since $v_\theta(s, \tau, l)$ represents a *policy-marginalized* transport direction, we align it with the expected drift under the current policy, approximated by the replay sample:

$$s_t^{\text{flow}}(\kappa) = s_t + \kappa\, v_\theta(s_t, \tau, l), \tag{18}$$

where $\kappa > 0$ maps a small planning-time step to a single environment-time transition (we set $\kappa = \Delta\tau$). We minimize

$$\mathcal{L}_{\text{cons}}(\theta) = \mathbb{E}\left[\|\Delta s_t - \kappa\, v_\theta(s_t, \tau, l)\|_2^2\right]. \tag{19}$$

In practice, the expectation is taken over replay samples $(s_t, a_t, l)$ (and optionally $\tau \sim \mathcal{U}[0, 1]$), linking the planning-time vector field to environment-time dynamics.

---

**Algorithm 1** Practical FlowMAP Implementation

1: Initialize policy $\pi_\theta$, critic $\hat{V}_\psi$, encoder producing $s_t$, flow field $v_\theta$, and replay buffer $\mathcal{D}$
2: **for** each training iteration **do**
3:     Roll out $\pi_\theta$; infer $s_t$; store transitions in $\mathcal{D}$
4:     Sample a batch from $\mathcal{D}$ and form the current meta-distribution $\rho_0^l := \rho_t^l$ in (3)
5:     Build value-shaped target $\rho_\star^l$ from (10) using $\hat{V}_\psi$
6:     Compute coupling $\eta^l \in \Pi(\rho_0^l, \rho_\star^l)$ via (11); sample endpoints $(s^{(0)}, s^{(1)}) \sim \eta^l$
7:     Sample $\tau \sim \mathcal{U}[0, 1]$; construct $s_\tau$ and $v^{\text{ref}}$
8:     Compute flow loss $\mathcal{L}_{\text{FM}}$ in (13)
9:     Compute consistency loss $\mathcal{L}_{\text{cons}}$ in (19)
10:    Update $(\theta, \psi)$ by minimizing $\mathcal{L}_{\text{total}}$ in (20)
11: **end for**

---

FPCT optimizes a standard actor–critic objective augmented by distributional planning and consistency. The key rationale is that the three components address complementary requirements for end-to-end control under dynamic heterogeneity: $\mathcal{L}_{\text{RL}}$ supplies the training signal from sparse, delayed rewards; $\mathcal{L}_{\text{FM}}$ provides a distribution-level planning objective that explicitly reshapes meta-state occupancy toward the value-shaped target $\rho_\star^l$; and $\mathcal{L}_{\text{cons}}$ ties the planning-time transport directions to environment-feasible transitions, preventing the learned flow from drifting into unrealizable directions in $\mathcal{S}$. Optimizing them jointly therefore couples "what to achieve", "where to move probability mass", and "how to remain realizable", yielding a single training signal that co-adapts the representation, flow, and policy. Formally, FlowMAP minimizes

$$\mathcal{L}_{\text{total}}(\theta) = \mathcal{L}_{\text{RL}}(\theta) + \lambda_{\text{FM}}\mathcal{L}_{\text{FM}}(\theta) + \lambda_{\text{cons}}\mathcal{L}_{\text{cons}}(\theta), \tag{20}$$

where $\mathcal{L}_{\text{RL}}$ is the backbone RL loss, and $\lambda_{\text{FM}}, \lambda_{\text{cons}} \geq 0$ balance return optimization, distributional transport, and realizability. This joint training makes the learned transport in $\mathcal{S}$ directly regularize policy learning, so that the policy-induced meta-distribution dynamics better match the intended value-transport under dynamic heterogeneity.

## 3.4. Practical Implementation of FlowMAP

Algorithm 1 summarizes a minimal implementation of FlowMAP. In practice, $\rho_0^l$ and $\rho_\star^l$ are not explicitly parameterized: $\rho_0^l$ is approximated by sampled meta-states from replay, while $\rho_\star^l$ is approximated by a value-preferred target pool selected according to $\hat{V}_\psi$. Each update jointly optimizes the actor–critic loss together with value-transport flow matching (13) and the realizability constraint (19), yielding the total objective (20). Overall, FlowMAP converts value estimates into target-driven transport over empirical meta-state occupancies, improving robustness beyond step-wise updates under dynamic heterogeneity.

*Table 1.* Benchmark protocol and training configurations. We report the interaction budget in *environment steps*, the action repeat, and the number of parallel environment instances.

| Benchmark | Interaction budget | Action repeat | # Env instances |
|---|---|---|---|
| Minecraft (Diamond) (Hafner, 2021) | 100M | 1 | 64 |
| DMLab-30 (Guo et al., 2020) | 100M | 4 | 16 |
| ProcGen (Cobbe et al., 2020) | 50M | 1 | 16 |
| Atari (Mnih, 2013) | 200M | 4 | 16 |
| Atari100k (Ye et al., 2021) | 400K (=100K agent steps) | 4 | 1 |
| Visual Control (Tassa et al., 2018) | 1M | 2 | 16 |
| Proprioceptive Control (Tassa et al., 2018) | 500K | 2 | 16 |
| BSuite (Osband et al., 2020) | Prescribed protocol | – | 1 |

## 4. Experiments

We evaluate FlowMAP for environment-generalizable agent planning under *dynamic heterogeneity*. Concretely, we ask whether learning to transport meta-state distributions in the planning space $\mathcal{S}$ (i) improves generalization to task-conditioned, nonstationary shifts, and (ii) provides a scalable planning signal when rewards are sparse and delayed.

### 4.1. Experimental Settings

Across all benchmarks, we follow the standard environment configurations for each suite. Table 1 summarizes the interaction budgets, action repeats, and the number of parallel environment instances used for training. The concrete grounding of abstract symbols (e.g., $s_t$, $\xi_l$, and $\rho_t^l$) in each benchmark is summarized in Appendix (Table 5).

**Compute and seeds.** Unless noted otherwise, each agent is trained on a single NVIDIA A6000 GPU. We run 5 random seeds per benchmark by default; due to benchmark-specific requirements and computational constraints, we run 10 seeds for BSuite and 10 seeds for Minecraft to reliably estimate the fraction of runs that achieve diamonds.

**Evaluation metrics.** Unless otherwise specified by the benchmark convention, we report **episodic return**, i.e., the undiscounted sum of environment rewards over an episode. For Atari-style aggregate comparisons, we follow the standard normalized-score reporting convention used by the corresponding benchmark.

**Baselines.** We compare FlowMAP against strong, widely used baselines that represent dominant paradigms across the same benchmark suite. These include a state-of-the-art world-model agent (**DreamerV3** (Hafner et al., 2025)) and established model-free baselines such as **PPO** (Schulman et al., 2017), **Rainbow** (Hessel et al., 2018), and **IMPALA** (Espeholt et al., 2018), as well as the standard low-data Atari100k baselines **SimPLe** (Kaiser et al., 2020), **Efficient-MuZero** (Ye et al., 2021), **SPR** (Schwarzer et al., 2021), **IRIS** (Micheli et al., 2023), and **TWM** (Robine et al., 2023).

All baselines are evaluated under the same benchmark protocols and reporting conventions as FlowMAP.

### 4.2. Experimental Analysis

Figure 3 reports the performance of FlowMAP across eight benchmark families, covering diagnostic tasks, procedurally generated environments, long-horizon visual domains, and open-world control settings. Across the benchmark suite, FlowMAP is competitive with or outperforms strong baselines such as Dreamer. The gains are most evident in environments with pronounced dynamic heterogeneity, such as ProcGen and Minecraft, where procedural layouts, latent configuration variables, and emergent subgoal structures induce large variation across episodes. In these settings, fixed Markov state representations or explicit environment models are brittle, since identical observations and actions may correspond to different latent regimes. By operating on meta-state occupancy distributions, FlowMAP directly shapes how probability mass over internal hypotheses redistributes under shifts, leading to more stable learning across heterogeneous episodes. On data-limited and sparse-reward benchmarks such as Atari100k and BSuite, FlowMAP also shows improved robustness over step-wise value optimization methods. Rather than relying on a well-calibrated critic, Value-Transport Flow Matching (VTFM) uses value estimates only to define a soft target distribution in meta-state space, transforming noisy scalar signals into a distribution-level planning objective. This reduces sensitivity to value miscalibration under shift and stabilizes long-horizon learning. Moreover, Flow–Policy Co-Training (FPCT) aligns planning-time transport with environment-feasible dynamics, ensuring that distributional control corresponds to realizable behavior in partially observable domains.

Overall, the results support the central claim of FlowMAP: robust planning under dynamic heterogeneity arises not from explicit latent modeling or accurate long-horizon rollouts, but from converting value signals into distribution-level control objectives and aligning planning-time transport with policy learning and environment dynamics.

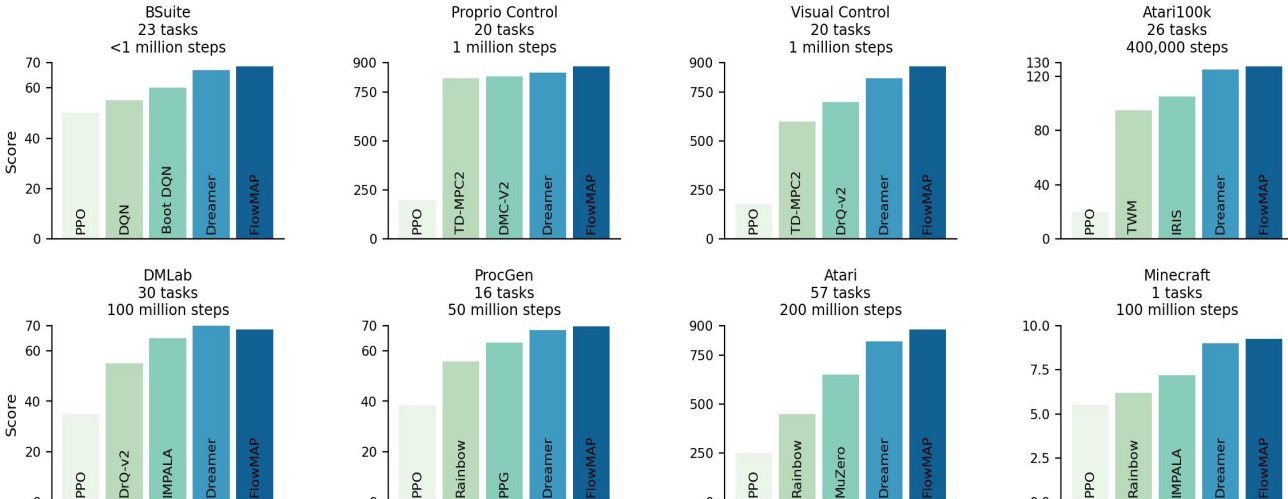

*Figure 3.* **Main results across benchmarks.** We report aggregate task performance for FlowMAP and strong baselines over eight benchmark families spanning diagnostic suites, procedurally generated environments, long-horizon open-world tasks, and pixel/state-control domains. All methods are trained and evaluated under the same interaction budgets and protocols in Table 1, and we follow the standard reporting conventions of each benchmark. While the main figures focus on final performance, FlowMAP also demonstrates improved learning efficiency, reaching strong performance substantially earlier under the same interaction budgets (Appendix C).

### 4.3. Hyperparameter Sensitivity

We further examine the sensitivity of FlowMAP on Minecraft to two key hyperparameters in VTFM: the quantile filtering ratio $\alpha$ for constructing the value-preferred target pool, and the temperature parameter $\beta$ for shaping the value-induced target distribution. Figure 4 shows that performance on Minecraft is strongest around $\alpha = 0.005$ ($9.0 \pm 0.3$). Very small values make the target pool overly selective and less stable, while larger values weaken filtering by admitting more low-value samples, as reflected by the decline at $\alpha = 0.05$ and $\alpha = 0.1$. Figure 5 shows a similar intermediate optimum for $\beta$: performance peaks around $\beta = 1$ ($9.0 \pm 0.3$), whereas both overly small and overly large values reduce return. These results confirm that FlowMAP is robust to hyperparameter variation and performs best under a stable intermediate value-shaping regime.

### 4.4. Ablation Study

To isolate the contribution of FlowMAP components, we conduct ablation studies on both **Minecraft** and **BSuite**. Minecraft is an open-world benchmark with long horizons and pronounced dynamic heterogeneity from procedural world generation, while BSuite provides a diagnostic suite for evaluating whether the component-wise trends persist beyond a single open-world domain. For each variant, we remove or modify one component while keeping the backbone architecture, training budget, and hyperparameters fixed. All variants share the same meta-state encoder, action space, and replay buffer. We evaluate the following variants:

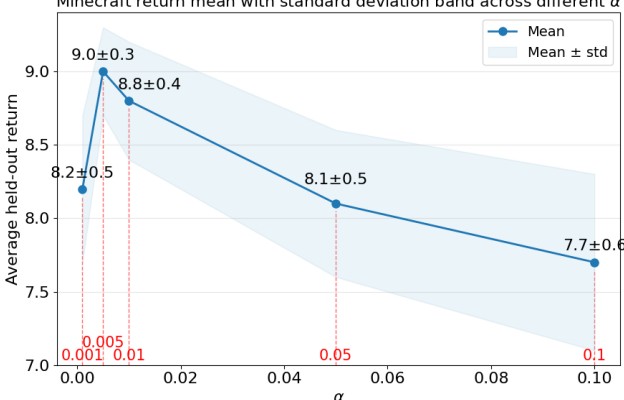

*Figure 4.* Minecraft episodic return under different values of the quantile filtering ratio $\alpha$. Performance is strongest near $\alpha = 0.005$: very small values make the target pool overly selective, while larger values weaken filtering by admitting more low-value samples.

**No Flow** (actor–critic without planning-time flow), **RL + Cons only** (actor–critic with the consistency regularizer but without value-transport flow matching), **Flow w/o Value Target** (flow matching with a uniform target), **Pointwise Value Update** (replacing distributional transport by local ascent of $\nabla_s \hat{V}(s, l)$), **VTFM w/o Consistency** (removing $\mathcal{L}_{\text{cons}}$), **Frozen Flow** (pretrained and frozen $v_\theta$), and **Full FlowMAP**. For Minecraft, all models are trained for the same number of environment steps and evaluated on held-out world seeds. For Minecraft, we report mean episode return over 5 random seeds with standard deviations in Table 2. For BSuite, we report mean episodic return in Table 3, since its diagnostic protocol exhibits low variance.

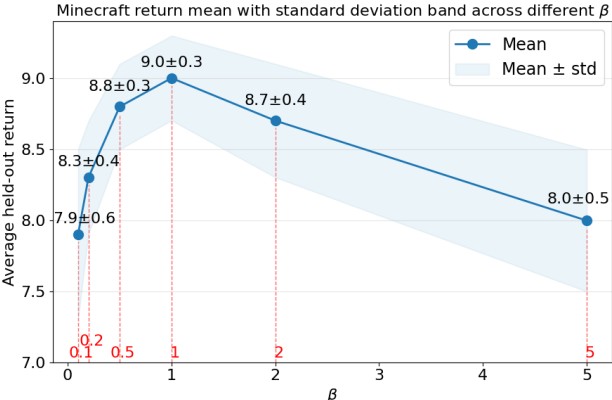

*Figure 5.* Minecraft episodic return under different values of the temperature parameter $\beta$. Performance is strongest around $\beta = 1$, while both smaller and larger values reduce return, indicating a stable intermediate regime.

*Table 2.* Ablation study on Minecraft. Each variant removes or modifies one component of FlowMAP. Results are averaged over 5 random seeds (mean $\pm$ std).

| Method | Minecraft Return |
|---|---|
| No Flow (RL only) | $6.0 \pm 0.4$ |
| RL + Cons only | $6.2 \pm 0.4$ |
| Flow w/o Value Target | $6.5 \pm 0.5$ |
| Pointwise Value Update | $6.7 \pm 0.4$ |
| VTFM w/o Consistency | $7.1 \pm 0.6$ |
| Frozen Flow | $7.6 \pm 0.3$ |
| **Full FlowMAP** | $\mathbf{9.0 \pm 0.2}$ |

*Table 3.* Ablation study on BSuite. Values are reported as mean episodic return averaged over tasks. Because BSuite is a diagnostic benchmark and shows low evaluation variance in our runs, we report mean returns for compactness. The full method performs best, while removing key components reduces performance.

| Method | BSuite Mean Return |
|---|---|
| No Flow (RL only) | 54.2 |
| RL + Cons only | 55.5 |
| Flow w/o Value Target | 53.8 |
| Pointwise Value Update | 61.3 |
| VTFM w/o Consistency | 57.8 |
| Frozen Flow | 65.2 |
| **Full FlowMAP** | **69.0** |

The ablation results on Minecraft and BSuite consistently support the contribution of each FlowMAP component. On Minecraft, removing the planning-time flow substantially degrades performance, showing that step-wise policy/value learning over meta-states is insufficient under strong heterogeneity. Adding the consistency term alone brings only a small improvement over the RL-only baseline, indicating that realizability alignment is useful but cannot replace value-directed distributional transport. Flow without value shaping yields limited gains, and pointwise value ascent also underperforms distributional transport, suggesting that effective planning requires a target-aware redistribution of meta-state occupancy rather than local value-gradient updates. Removing the consistency term increases variance and lowers return, consistent with transport directions drifting away from feasible action-induced dynamics. The BSuite results exhibit the same component-wise trend: the full method achieves the highest task-averaged return, while variants that remove value-shaped transport, consistency alignment, or adaptive flow–policy co-training all perform worse. In particular, the gap between **Frozen Flow** and **Full FlowMAP** shows that a fixed transport field is less effective than one that co-adapts with the policy and representation. Together, these results validate the necessity of coupled design.

Overall, these ablations show that FlowMAP's gains arise from the joint effect of value-shaped target construction, distributional transport, and feasibility alignment, rather than from any single auxiliary loss.

## 5. Conclusion

We studied agent planning under *dynamic heterogeneity*, where observations and dynamics shift. We proposed **FlowMAP**, which frames open-world planning as continuous-time flow matching over meta-state occupan-

cies by learning a task-conditioned velocity field that transports the current meta-state distribution toward a target. FlowMAP further introduces Value-Transport Flow Matching (VTFM) to turn value estimates into a value-shaped target distribution, and Flow–Policy Co-Training (FPCT) to align the learned transport with action-induced dynamics via joint optimization with shared representations. Experiments across diverse benchmarks show that FlowMAP is competitive with strong model-free and world-model baselines.

We hope FlowMAP can inspire the agent-planning community to view distribution-level control over learned internal states as a useful primitive for robust planning under environmental shifts and long-horizon uncertainty. In future work, we plan to incorporate mean-flow-style objectives to obtain smoother velocity estimation and reduce variance in planning-time transport. This may further improve the stability and performance of FlowMAP in long-horizon environments with sparse rewards and strong dynamic heterogeneity. More broadly, as agents increasingly interact with complex real-world environments, we hope FlowMAP provides a small but meaningful example of how science-inspired principles—such as transport, dynamics, and distributional control—can offer reliable foundations for building more generalizable AI systems.

## Acknowledgements

This work was supported by the Joint Funds of the National Natural Science Foundation of China under (Grant No. U22A2099 and U2441242), the General Program of the National Natural Science Foundation of China under (Grant No. 62376028), the National Key Research and Development Program of China under (Grant No. 2025YFC3309100), the Excellent Young Scientists Fund (Overseas) of the National Natural Science Foundation of China, the National Key Scientific Instruments and Equipment Development Project under (Grant No. 62427808).

## Impact Statement

This paper presents work whose goal is to advance the field of Machine Learning. There are many potential societal consequences of our work, none of which we feel must be specifically highlighted here.

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

# A. Discussion of Related Work

## A.1. Positioning Summary

Many flow-based RL variants reshape a reward-induced target over actions or terminal outcomes. In contrast, FlowMAP explicitly learns a planning-time velocity field that transports the meta-state occupancy distribution toward a value-shaped target distribution, and further constrains the transport to be consistent with environment interaction.

## What is new in FlowMAP.

- **Distribution-level planning signal under dynamic heterogeneity:** FlowMAP formulates planning as continuous-time transport of the meta-state occupancy distribution.

- **Value-Transport Flow Matching (VTFM):** We construct a value-shaped target distribution and an efficient endpoint coupling to provide a stable planning-time supervision signal even with sparse and delayed rewards.

- **Flow–Policy Co-Training (FPCT):** We co-optimize the transport dynamics and the policy to align the learned flow with action-environment interaction, reducing unrealizable transport directions.

## A.2. Discussion of LLM-Based Agent Planning

Recent work has explored large language models (LLMs) as central reasoning or planning components in agent systems (Xie et al., 2024; Xiong et al., 2026). While these approaches have shown promise in instruction following, task decomposition, and symbolic plan synthesis, this paper deliberately does not consider LLM-centric agent planning as its primary focus (Valmeekam et al., 2023; Kambhampati et al., 2024; Xiong et al., 2025).

**Optimization objective mismatch.** At a high level, reinforcement-learning-based agent planning and LLM-based planning optimize fundamentally different objectives. A standard RL agent seeks to learn a policy

$$\pi^* = \arg\max_{\pi} \mathbb{E}_{\pi}\left[\sum_{t=0}^{\infty} \gamma^t r_t\right], \qquad (21)$$

where the expectation is taken over environment trajectories induced by interaction, and optimality is defined with respect to long-horizon return under environment dynamics.

In contrast, an LLM used as a planner typically generates an action sequence or plan

$$(a_0, a_1, \ldots, a_T) \sim p_{\theta}(\cdot \mid \text{prompt}), \qquad (22)$$

by maximizing next-token likelihood under a static language modeling objective. Even when augmented with feedback

or re-prompting, the LLM does not directly optimize (21) through interaction-driven credit assignment, nor does it maintain an explicit representation of policy-induced state distributions.

**Distributional state control vs. sequence generation.** A key distinction arises under partial observability and dynamic heterogeneity. RL-based planning implicitly or explicitly reasons over *state or belief distributions*. In this work, this is captured by the meta-state occupancy distribution

$$\rho_t^l(s) = p(s_t = s \mid \pi, l), \qquad (23)$$

whose evolution depends jointly on environment dynamics, observation processes, and the policy.

LLM-based planning, by contrast, operates primarily at the level of *symbolic or textual action sequences*. It does not explicitly control how probability mass over internal hypotheses or latent regimes evolves under execution. As a result, when environment dynamics shift or observations alias across regimes, there is no principled mechanism for redistributing belief mass or correcting long-horizon plans beyond ad hoc re-prompting.

**Sensitivity under long horizons and execution noise.** In long-horizon tasks, planning errors compound through execution. For model-based RL, this manifests as model bias or rollout drift, which can be mitigated by value grounding or policy learning. For LLM-based planners, however, execution feedback is weakly coupled to generation: the planner is not updated by gradients that reflect downstream execution failure. Formally, the plan generator $p_{\theta}$ is optimized offline and remains fixed during deployment, so planning errors under regime shifts cannot be corrected by minimizing a trajectory-level loss such as

$$\mathcal{L}_{\text{exec}} = \mathbb{E}\left[\sum_{t} \ell(x_t, a_t)\right], \qquad (24)$$

where $\ell$ reflects deviation from feasible or high-value behavior.

**Role of LLMs as auxiliary components.** These limitations do not imply that LLMs are uninformative for agent systems. Rather, they suggest that LLMs are better suited as *auxiliary modules*—for example, providing semantic priors, task abstractions, or heuristic guidance—while the core planning mechanism remains grounded in reinforcement learning and control. This view aligns with prior critiques in the automated planning and robotics communities, which caution against treating LLMs as drop-in replacements for planners or controllers, especially in long-horizon, stochastic, and partially observable environments.

| Paradigm | Primary control object | Training signal / target | Shift-aware | Needs env model | Feasibility alignment |
|---|---|---|---|---|---|
| **FlowMAP (Ours)** | Meta-state occupancy transport | Value-shaped *target distribution* + flow matching loss | ✓ | No explicit rollout planner | ✓(FPCT) |
| GFlowNet / FlowRL / Rein-Flow | Action / terminal outcome distribution | Reward-shaped terminal/outcome/action targets | Partial | No | No / implicit |
| Diffusion / flow-based planning | Plan / trajectory samples | Denoising / score matching on plans (often offline / model-based) | Partial | Often | Varies |
| Occupancy / OT control | State(-action) occupancy optimization | OT / occupancy objectives, policy optimization | Partial | No | Implicit |
| World-model planning (e.g., latent imagination) | Imagined trajectories / latent rollouts | Model learning + value/policy optimization | Partial | ✓ | Implicit |
| Distributional RL (e.g., return distributions) | Return distribution at a state-action | Bellman-style distributional targets | No (by default) | No | No |

*Table 4.* High-level positioning of FlowMAP against adjacent lines of work. "Shift-aware" indicates whether the method is explicitly designed to address distribution shift induced by dynamic heterogeneity.

**Position of FlowMAP.** The objective of this paper is therefore not to replace language-based reasoning, but to strengthen the planning backbone of RL agents themselves. FlowMAP addresses dynamic heterogeneity by explicitly shaping how policy-induced meta-state distributions evolve over time, a form of distribution-level control that is not naturally expressed by sequence-generation-based planners. For this reason, we restrict our attention to RL-based agent planning and leave systematic integration with LLM-assisted planning to future work.

# B. Sources of Dynamic Heterogeneity Across Benchmarks

## B.1. Dynamic Heterogeneity Across Benchmarks

This appendix concretizes the latent heterogeneity factor $\xi_l \in \Xi$ in our experimental domains. Across open-world benchmarks, *dynamic heterogeneity* arises from a mixture of (i) **unobserved configuration variables** (e.g., seeds, layouts, object/agent states, hidden parameters), (ii) **task-conditioned rules/objectives** (e.g., level-specific reward logic, termination conditions, action semantics), and (iii) **nonstationary evolution** of the environment and the agent–environment interaction over time (e.g., phase changes, inventory progression, difficulty schedules). In all cases, $\xi_l$ is typically high-dimensional, only partially expressed through observations and transitions, and weakly supervised

by sparse rewards, making explicit modeling difficult: $\xi_l$ is not directly labeled, it couples with both $F(\cdot \mid x, a, \xi_l)$ and $O(\cdot \mid x, a, \xi_l)$, and it can change the effective Markov structure under partial observability and action repetition.

Below we detail, for each benchmark family, (a) where heterogeneity comes from, (b) how it manifests in observations/dynamics/objectives, and (c) why it is hard to model explicitly. For clarity, Table 5 summarizes how the abstract symbols used throughout the paper (e.g., $s_t$, $\xi_l$, and $\rho_t^l$) are concretely instantiated across different benchmark families.

### B.1.1. MINECRAFT

**Dynamic heterogeneity in Minecraft.** Minecraft-style open-world tasks exhibit heterogeneity that is simultaneously procedural, compositional, and long-horizon, making the effective environment regime vary substantially across episodes and over time. The latent factor $\xi_l$ encompasses procedural world generation (e.g., seeds, biomes, terrain topology, and resource distributions), hidden entity and object configurations that are only partially observable from egocentric views, and progress-dependent interaction rules induced by inventory and tool states. In addition, stochastic events such as entity behavior and combat outcomes further alter the effective transition dynamics.

These sources manifest as strong nonstationarity in both observations and dynamics. Visually similar observations

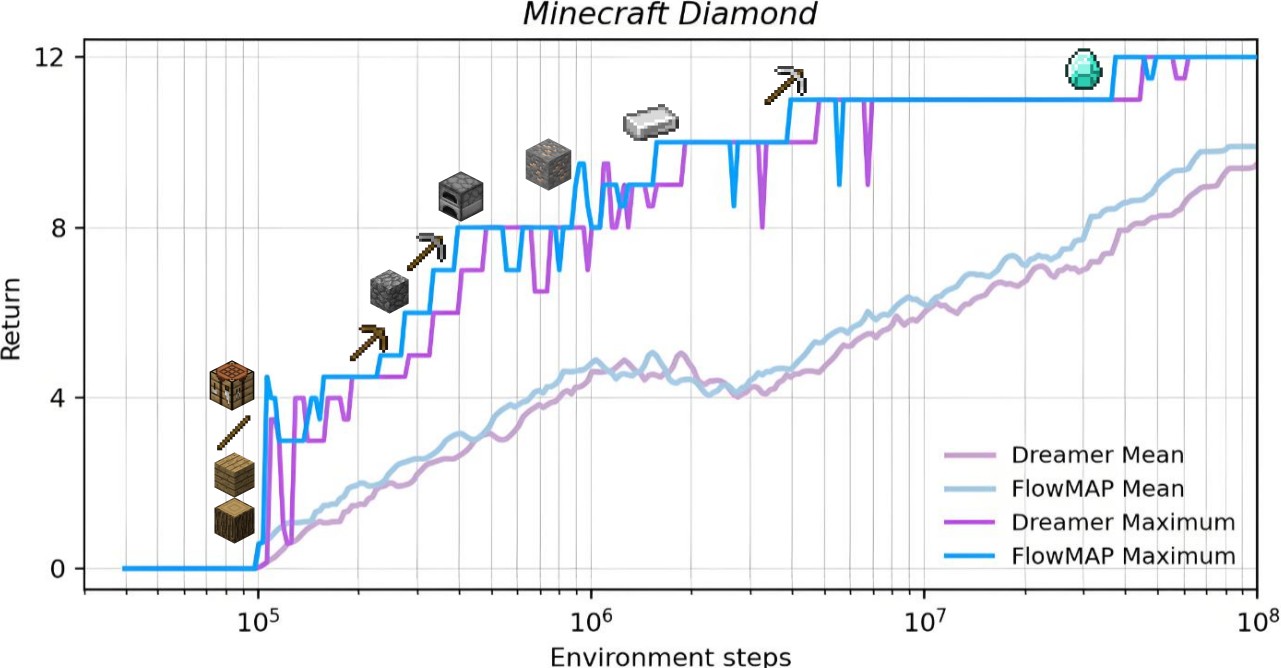

*Figure 6.* Learning curve on Minecraft Diamond under a fixed interaction budget. Although FlowMAP and Dreamer reach similar final returns, FlowMAP converges faster, indicating improved learning efficiency in long-horizon sparse-reward planning.

can correspond to distinct underlying configurations due to occlusion, lighting, or spatial layout, while identical actions may induce qualitatively different transitions depending on local geometry, nearby entities, or inventory-dependent affordances. Objectives also evolve over long horizons, as newly crafted tools unlock transitions that were previously infeasible, reshaping the space of high-reward behaviors. Sparse and delayed rewards exacerbate credit assignment, providing only weak supervision for inferring $\xi_l$.

Explicitly modeling this heterogeneity is difficult. The latent space induced by procedural generation, spatial structure, and progress-dependent rules is combinatorial and high-dimensional, with no direct supervision. Under partial observability, many causal determinants of transitions and rewards remain unobserved, and long-horizon dependencies entangle $\xi_l$ with the policy's own visitation distribution, introducing selection bias. These properties make parametric modeling of $\xi_l$ brittle in practice.

### B.1.2. DMLAB-30

**Dynamic heterogeneity in DMLab-30.** DMLab-30 combines substantial inter-task diversity with within-task heterogeneity arising from procedural episode generation. The latent factor $\xi_l$ includes task identity and rule sets, randomized map layouts, spawn configurations, and stochastic entity behaviors. Action repetition and frame skipping further

interact with partial observability, modifying the effective dynamics experienced by the agent.

Heterogeneity appears as shifts in observation distributions across tasks and episode seeds, as well as dynamics variation where identical actions may be safe or catastrophic depending on the layout. Reward structure and success criteria differ markedly across tasks, leading to task-dependent credit assignment profiles and temporal nonstationarity within episodes.

Explicit modeling is challenging because $\xi_l$ mixes discrete variables (task identity, rules) with high-dimensional continuous factors (map configurations), all only partially observed from egocentric views. Reward sparsity and delay obscure which latent factors are causally responsible for success, while action repetition aggregates over unobserved intermediate transitions, increasing perceptual aliasing.

### B.1.3. PROCGEN

**Dynamic heterogeneity in ProcGen.** ProcGen benchmarks are explicitly designed to test generalization across procedural seeds, inducing controlled but systematic distribution shifts. Here, $\xi_l$ primarily reflects the latent state of the level generator, determining layouts, obstacles, enemies, and reward placement, along with stochastic local interactions. This heterogeneity manifests as seed-induced shifts in both observations and dynamics: training and eval-

uation seeds yield distinct state-action distributions, and action effects depend on procedurally generated local geometry. Reward timing and location vary with level structure, complicating temporal credit assignment.

Explicit modeling would require learning an implicit model of the generator itself, a nonlinear and environment-specific process. Partial observability further limits inference of global layout from local views, making accurate estimation of $\xi_l$ difficult under sparse rewards.

### B.1.4. ATARI AND ATARI100K

**Dynamic heterogeneity in Atari benchmarks.** Atari benchmarks exhibit pronounced inter-game heterogeneity, as each game defines distinct dynamics, visuals, and reward semantics, as well as within-game nonstationarity through phase changes, difficulty schedules, and stochastic opponents. Action repeat and frame skipping further alter the effective transition kernel. Heterogeneity appears as shifts in reward structure, regime changes within episodes, and representation mismatch when pixel observations entangle multiple objects whose relevance changes over time. In low-data settings such as Atari100k, these effects are amplified by limited experience.

Explicit modeling is hindered by the absence of annotations for hidden phase variables and game-specific mechanisms. Under data scarcity, small modeling errors can compound rapidly over long rollouts, undermining planning reliability.

### B.1.5. VISUAL CONTROL

**Dynamic heterogeneity in visual control tasks.** Pixel-based control tasks combine heterogeneity in rendering and latent physical parameters. The latent factor $\xi_l$ includes camera viewpoint, lighting, textures, occlusions, as well as unobserved physical properties such as friction, damping, and contact conditions. This leads to observation shifts where identical physical states produce different images, and to dynamics sensitivity in contact-rich regimes where small latent differences yield discontinuous transitions. Sparse early rewards provide limited guidance for disentangling these factors.

Explicit modeling from pixels is statistically demanding: rendering and physics are entangled, depth and contact modes are not uniquely identifiable, and learned dynamics models are brittle under long-horizon compounding error.

### B.1.6. PROPRIOCEPTIVE CONTROL

**Dynamic heterogeneity in proprioceptive control.** Even with state observations, heterogeneity arises from unobserved physical parameters, stochastic disturbances, and task-dependent reward sensitivities. Hidden factors such as friction, masses, and actuator noise alter effective dynamics,

while contact regimes can switch within an episode.

Identifying these latent parameters requires sufficient excitation and coverage, which conflicts with sparse-reward exploration. Multiple latent configurations can produce similar short-horizon trajectories, rendering regime inference ambiguous.

### B.1.7. BSUITE

**Dynamic heterogeneity in BSuite.** BSuite primarily reflects heterogeneity at the task and protocol level. The latent factor $\xi_l$ corresponds to task identity and configuration, determining observation structure, reward delay, and success criteria.

This induces strong variation in learning dynamics and failure modes across tasks. Explicit task-specific modeling would defeat the diagnostic purpose of the suite; robustness must instead arise from learning dynamics and representation-level adaptation.

## C. Additional Experiment

In addition to final performance, we analyze the *learning efficiency* of FlowMAP on Minecraft Diamond by examining how quickly agents reach strong task performance under a fixed interaction budget. Figure 6 reports the training curve on Minecraft Diamond, where FlowMAP and Dreamer ultimately achieve comparable final scores. Although final returns are similar, FlowMAP reaches high-performing regimes substantially earlier. This gap is particularly visible in this long-horizon and sparse-reward environment, where exploration and credit assignment dominate early learning dynamics. In this setting, FlowMAP exhibits faster early-stage improvement and reduced variance across seeds, indicating more stable optimization. We attribute this behavior to the distribution-level planning signal introduced by Value-Transport Flow Matching (VTFM). By reshaping the meta-state occupancy distribution toward value-preferred regions, FlowMAP provides structured guidance even when scalar rewards are delayed or noisy. As a result, the policy is exposed earlier to productive regions of the state space, accelerating policy improvement without relying on long-horizon rollouts.

Notably, this improvement in convergence speed does not come at the expense of asymptotic performance. FlowMAP maintains competitive or superior final scores while achieving them with fewer effective interactions, suggesting that distribution-level transport acts as a complementary planning regularizer that improves learning efficiency rather than merely shifting the final optimum.

Table 5. Grounding of abstract symbols across benchmark families.

| Benchmark | Symbol | Concrete meaning |
|---|---|---|
| Minecraft | $x_t$ | Full unobserved world state, including terrain, blocks, entities, and inventory. |
| | $o_t$ | Egocentric RGB observation with occlusion and limited field-of-view. |
| | $a_t$ | Movement, mining, crafting, placement, and interaction actions. |
| | $s_t$ | Meta-state encoding inferred world structure and long-horizon hypotheses. |
| | $\rho_t^l$ | Distribution over inferred world hypotheses induced by policy interaction. |
| | $\hat{V}(s,l)$ | Estimated long-horizon return (e.g., progress toward diamond). |
| DMLab-30 | $x_t$ | Simulator state including agent pose, map layout, objects, and task variables. |
| | $o_t$ | Egocentric rendered observation. |
| | $a_t$ | Discrete navigation and interaction actions with action repeat. |
| | $s_t$ | Meta-state summarizing task identity, map configuration, and progress. |
| | $\rho_t^l$ | Distribution over task- and layout-specific meta-states. |
| | $\hat{V}(s,l)$ | Task-conditioned value estimate (navigation, collection, etc.). |
| ProcGen | $x_t$ | True game state including latent level generator variables. |
| | $o_t$ | Local visual observation of procedurally generated geometry. |
| | $a_t$ | Discrete control actions. |
| | $s_t$ | Meta-state encoding inferred level structure from partial views. |
| | $\rho_t^l$ | Occupancy over inferred level hypotheses under interaction. |
| | $\hat{V}(s,l)$ | Estimated return conditioned on inferred level layout. |
| Atari / Atari100k | $x_t$ | Emulator state including RAM, hidden counters, and phase variables. |
| | $o_t$ | Pixel observations after preprocessing and frame stacking. |
| | $a_t$ | Discrete joystick/button actions with frame skip. |
| | $s_t$ | Latent or meta-state summarizing temporal context. |
| | $\rho_t^l$ | Meta-state occupancy induced by interaction within a game. |
| | $\hat{V}(s,l)$ | Estimated discounted return under current game regime. |
| Visual Control | $x_t$ | Underlying physical simulator state (positions, velocities, contacts). |
| | $o_t$ | Rendered pixel observation affected by camera and lighting. |
| | $a_t$ | Continuous control actions. |
| | $s_t$ | Meta-state encoding inferred physical configuration from pixels. |
| | $\rho_t^l$ | Distribution over inferred physical regimes. |
| | $\hat{V}(s,l)$ | Value estimate under inferred dynamics. |
| Proprioceptive Control | $x_t$ | True simulator state including hidden physical parameters. |
| | $o_t$ | Proprioceptive observations (joint angles, velocities). |
| | $a_t$ | Continuous motor commands. |
| | $s_t$ | Meta-state encoding inferred latent dynamics and regime switches. |
| | $\rho_t^l$ | Occupancy over inferred dynamic regimes. |
| | $\hat{V}(s,l)$ | Estimated return under current inferred dynamics. |
| BSuite | $x_t$ | Task-specific environment state. |
| | $o_t$ | Diagnostic observation defined by each task. |
| | $a_t$ | Discrete task-dependent actions. |
| | $s_t$ | Meta-state summarizing inferred task structure and memory. |
| | $\rho_t^l$ | Distribution over task-conditioned meta-states. |
| | $\hat{V}(s,l)$ | Task-specific value estimate reflecting success criteria. |

