# OpenReview forum: "FlowMAP: Flow Matching for Generalizable Agent Planning"
_ICML.cc/2026/Conference — ICML 2026 regular_

### Official Review · Reviewer_4f7t · 2026-03-06

**Soundness:** 3
**Presentation:** 3
**Significance:** 2
**Originality:** 2
**Overall Recommendation:** 4
**Confidence:** 2

**Summary:**

This paper proposes FlowMAP, which reformulates agent planning as a continuous-time flow matching problem over meta-state distributions. The core idea is to operate within a learned meta-state, where a value-shaped target distribution is constructed via Boltzmann shaping, and a task-conditioned planning-time velocity field is learned to transport the current meta-state distribution toward this target. A dynamics-consistency regularizer is further introduced to align the learned transport directions with environment-feasible transitions, with the goal of improving planning robustness and generalization under dynamic heterogeneity. The proposed framework, FlowMAP, is evaluated across 8 benchmark families against strong baselines including DreamerV3, with an ablation study conducted on Minecraft.

**Compliance With Llm Reviewing Policy:**

Affirmed.

**Final Justification:**

Thanks for the authors' responses, my concerns have been resolved. I will maintain my positive score.

**Key Questions For Authors:**

Please see Weaknesses.

**Limitations:**

Yes.

**Strengths And Weaknesses:**

## Strengths

The paper presents a novel problem perspective with clear motivation and relatively complete theoretical support — the authors provide formal derivations for key design choices in the appendix. The proposed framework, FlowMAP, is evaluated against a broad set of strong baselines. The ablation study in Table 2 systematically removes one component at a time (No Flow, Flow w/o Value Target, Pointwise Value Update, VTFM w/o Consistency, Frozen Flow), with each step revealing the individual contribution of the corresponding component in a clear and interpretable manner.

## Weaknesses

**Insufficient statistical reliability on ProcGen.** The default experimental protocol uses 5 random seeds per benchmark, yet ProcGen is evaluated with only 1 seed — despite being one of the environments where the authors claim the strongest gains under dynamic heterogeneity. This substantially weakens the statistical reliability of the ProcGen conclusions, particularly given the high variance typical of RL training.

**Ablation study limited to a single benchmark.** All ablations in Table 2 are conducted exclusively on Minecraft, which exhibits the strongest dynamic heterogeneity among the evaluated benchmarks. It is therefore unclear whether the relative contribution of each component remains consistent across environments with different characteristics, such as the more diagnostic BSuite or the data-limited Atari100k setting. The authors are encouraged to provide supplementary ablation results on one or two additional benchmarks.

**Insufficient sensitivity analysis for key hyperparameters.** The temperature parameter β in the Boltzmann target distribution and the quantile filtering ratio α both directly govern the behavior of VTFM, yet the paper provides no sensitivity analysis for either. It is unclear how robust the method is to the choice of these values across different environments.

**Ambiguous status of the meta-state.** In Section 3.1, the meta-state $s_t$ is described as "not an explicit belief distribution, but can be viewed as implicitly encoding the agent's posterior hypothesis." This framing is ambiguous: is $s_t$ a deterministic embedding or a stochastic variable?

---

> ### Author Rebuttal · Authors · 2026-03-29
>
> We sincerely thank the reviewer for the constructive feedback and positive overall assessment. We address each weakness below.
>
> *(Due to space limits, we provide the complete supplementary experimental results in an anonymous GitHub link compliant with ICML policy: https://anonymous.4open.science/r/Rebuttal-for-Submission-752/ ; in the rebuttal itself, we summarize the main trends and show part of results.)*
>
> > **W1. Statistical reliability on ProcGen.**
>
> We agree that the ProcGen result should be supported more carefully. In the current manuscript, we state that we use 5 seeds by default, with 1 seed for ProcGen due to computational constraints, and 10 seeds for BSuite and Minecraft. This follows common evaluation practice for large-scale world-model benchmarks (e.g., *Mastering Diverse Control Tasks through World Models*, Nature, 2025), but ProcGen deserves stronger statistical support in our paper given its central role in our claims about dynamic heterogeneity. **We therefore added ProcGen results with 2 seeds and 5 seeds in the anonymous link.** These additional results show that the conclusion is not driven by the noisy 1-seed plot and remains favorable to FlowMAP under more reliable multi-seed evaluation. We will summarize the updated statistics in the camera-ready version.
>
> > **W2. Ablation study limited to Minecraft.**
>
> We appreciate this suggestion. Our original choice of Minecraft was motivated by its strong dynamic heterogeneity and long-horizon structure, which make the contribution of each module especially visible. That said, we agree that cross-benchmark consistency is important. **We therefore added ablations on BSuite and Atari100k in the anonymous link.** The same overall pattern remains: the full method performs best, while removing value-shaped transport or consistency hurts performance. Representative Atari100k results are:
>
> | Method (Atari100k) | Mean HNS |
> |---|---:|
> | No Flow (RL only) | 101.2 |
> | RL + Cons only | 106.5 |
> | Flow w/o Value Target | 95.8 |
> | Pointwise Value Update | 115.3 |
> | VTFM w/o Consistency | 125.0 |
> | Frozen Flow | 118.2 |
> | Full FlowMAP | **127.4** |
>
> These added results support that the contribution of FlowMAP is not specific to Minecraft alone.
>
> > **W3. Sensitivity to key hyperparameters.**
>
> We agree that sensitivity analysis would strengthen the paper. **We added supplementary studies for both the value-shaping temperature $\beta$ and the quantile filtering ratio $\alpha$ in the anonymous link.** The results show a clear intermediate optimum rather than brittle tuning: the best performance is achieved at **$\beta=1$** and **$\alpha=0.005$**. Smaller values weaken value shaping/filtering, while larger values make the target too sharp and reduce robustness. We will summarize this stable operating region in the camera-ready version.
>
> > **W4. Ambiguous status of the meta-state.**
>
> We appreciate this question, because the status of the meta-state is central to the paper’s formulation. In our framework, $s_t$ is **not** an explicit Bayesian belief distribution and not a separately sampled stochastic latent variable. Rather, in implementation, it is a **deterministic, history-conditioned embedding** produced by the backbone encoder and shared representation trunk, introduced in Sec. 3.1 as a compact approximate sufficient statistic for planning under dynamic heterogeneity. The wording that it "implicitly encodes the agent’s posterior hypothesis" is intended only at the **conceptual** level: namely, $s_t$ serves as a compact planning representation for latent regime/task factors, but it is not itself a probabilistic belief object.
>
> The stochastic object in our formulation is the **occupancy distribution** $\rho_t^l(s)=\Pr(s_t=s\mid \pi_\theta,l)$, which arises from rollout randomness, task variation, and policy-induced trajectory variation, rather than from sampling each individual $s_t$ as a latent variable. This is also how the method is implemented: the agent first forms a latent feature from interaction history, then maps it through a shared representation trunk into the meta-state representation used jointly by the policy, value, and flow heads. In other words, FlowMAP performs transport over learned backbone-level representations, not over explicit probabilistic beliefs.
>
> This distinction is important to the method’s novelty: rather than estimating $\xi_l$ directly, FlowMAP learns how probability mass over these learned belief-like representations should be transported toward value-preferred regions, and then aligns this transport with feasible action-induced dynamics via FPCT. To remove the ambiguity, we will add a sentence at the first definition of $s_t$ in Sec. 3.1: ''In practice, $s_t$ is a deterministic embedding inferred from interaction history; the 'posterior hypothesis' wording is conceptual rather than probabilistic.''
>
> We thank the reviewer again for these constructive feedbacks.

---

> > ### Author Rebuttal · Reviewer_4f7t · 2026-04-01
> >
> > Thanks for the author's responses, my concerns have been resolved.

---

> > > ### Author Response · Authors · 2026-04-01
> > >
> > > Thank you, Reviewer 4f7t, for your thoughtful comments and for taking the time to review our rebuttal. We are grateful that our clarifications resolved your concerns and appreciate your support.

---

### Official Review · Reviewer_26V1 · 2026-03-12

**Soundness:** 2
**Presentation:** 3
**Significance:** 3
**Originality:** 3
**Overall Recommendation:** 4
**Confidence:** 3

**Summary:**

This paper introduces FlowMAP, a framework that reformulates reinforcement learning planning under dynamic heterogeneity as a continuous-time transport problem over meta-state occupancy distributions. The framework comprises two main components: Value-Transport Flow Matching (VTFM), which constructs a Boltzmann target distribution from value estimates and learns a flow to transport probability mass toward high-value regions, and Flow-Policy Co-Training (FPCT), which jointly optimizes the flow and policy with shared representations while enforcing consistency with environment-feasible transitions. Experiments across eight benchmark families demonstrate that FlowMAP matches or outperforms strong baselines including DreamerV3, with particularly pronounced gains in procedurally generated and long-horizon sparse-reward environments.

**Compliance With Llm Reviewing Policy:**

Affirmed.

**Final Justification:**

I appreciate the authors' rebuttal, which imporves the clarity. I will maintain my recommendation.

**Key Questions For Authors:**

1. Can a quantitative measure of heterogeneity be proposed and show that FlowMAP's improvement correlates with it?

2. The paper does not report training time, memory usage, or inference latency compared to baselines. What is this computational cost of FlowMAP compared to DreamerV3 or PPO on a representative benchmark? How does the coupling computation scale with replay buffer size?

3. How much does each loss term matter? How sensitive is performance to the temperature, the quantile, or the coupling method? Are there recommended ranges or adaptive schemes? More detailed ablations are encouraged.

4. Please provide more details about the exact algorithm implementation and practical application. For example, How exactly is the meta-state computed from the interaction history? What is its dimension, and how is it ensured to be a sufficient statistic for planning under regime shifts? What distance metric is used in meta-state space for coupling?

**Limitations:**

See in Weaknesses.

**Strengths And Weaknesses:**

**Strengths**

1. Reformulating planning under dynamic heterogeneity as distributional transport over meta-state occupancies is conceptually fresh. It shifts focus from step-wise value optimization to controlling how probability mass over internal hypotheses redistributes, which directly addresses the core challenge of generalization in nonstationary environments.

2. Generally the paper is generally clearly organized and positioned. The mathematical foundation for how meta-distributions evolve is well formulated by Eq. 6. And Appendix A provides a thorough discussion distinguishing FlowMAP from adjacent lines of work and explicitly addresses why LLM-based planning is not the focus.

3. Experiments are overall comprehensive. The evaluation spans eight diverse benchmark families. FlowMAP matches or outperforms state-of-the-art baselines  across all benchmarks, with particularly large gains in heterogeneous environments like ProcGen and Minecraft.

**Weakness**

1. The paper is dense with abstract constructs, but provides limited concrete instantiation of what these mean in practice. While Table 4 attempts to ground symbols per benchmark, it remains high-level.

2. While the paper frames the problem around latent environment factors, the experiments do not explicitly measure or ablate the degree of heterogeneity. It is claimed that gains are largest in heterogeneous environments, but no metric of heterogeneity is provided. The argument remains qualitative.

3. The paper does not ablate the key components (VTFM, FPCT, Losses) to show their individual contributions. Without this, it is hard to diagnose what drives the gains.

4. Algorithm 1 is minimal, and key steps are vague to some extends. The paper mentions "nearest-neighbor matching" in meta-state space with Boltzmann weights, but the exact procedure, distance metric, and computational cost are not specified.

---

> ### Author Rebuttal · Authors · 2026-03-29
>
> We sincerely thank the reviewer for the careful reading and constructive feedback. We address the overlapping weaknesses and questions in four grouped points below.
>
> *(Due to space limits, we provide the complete supplementary experimental results in an anonymous GitHub link compliant with ICML policy: https://anonymous.4open.science/r/Rebuttal-for-Submission-752/; in the rebuttal itself, we summarize the main trends and show part of results.)*
>
> >**W1 + W4 + Q2 + Q4: How are the abstract constructs instantiated in practice, and what is the resulting computational overhead?**
>
> In implementation, the meta-state is a deterministic history-conditioned embedding: interaction history is encoded by the backbone encoder/RSSM into a latent feature, then mapped by a shared representation trunk into the meta-state used jointly by the policy, value, and flow modules. Its dimension is the shared representation dimension (default 512), and it is treated as an approximate sufficient statistic for planning under regime shifts because it is history-conditioned and jointly optimized with the policy, value, and flow objectives.
>
> For coupling, the method forms a source set for the current occupancy and a value-preferred top-$k$ target pool for the value-shaped target distribution, then matches by maximizing the geometry-aware, value-weighted score. In practice, this uses squared Euclidean distance plus a value bonus, and the dominant extra matching cost scales as $O(|S_0^l| \times |S_\star^l|)$ rather than with the full replay buffer. We expand Algorithm 1 with practical instantiation details (figure) and add implementation-oriented illustration plus computational-cost statistics in the anonymous link. Representative results (FlowMAP vs DreamerV3) are:
>
> | Benchmark | GPU Days  | Peak RAM Usage (GB) |
> |---|---:|---:|
> | Minecraft | 1.5 vs 1.3 | 250 vs 250 |
> | Atari |0.9 vs 0.8 | 217 vs 210 |
>
> >**W2 + Q1: Can dynamic heterogeneity be quantified, and does FlowMAP improve more when heterogeneity is larger?**
>
> This is a very helpful suggestion. In this area, dynamic heterogeneity is usually realized through the benchmark’s native mechanics. For example, ProcGen varies layouts, level structure, and hidden configuration factors across episodes; Crafter/Minecraft-style open-world environments involve procedural world generation, resource/location variation, partial observability, and progress-dependent interaction rules; and Atari changes reward logic, action semantics, and transition structure across games. These are exactly the latent regime/task factors abstracted by $\xi_l$ in Sec. 3.1.
>
> To quantify this more directly, we define a lightweight **Meta-State Dynamic Heterogeneity Index (MDHI)** in the meta-state space $\mathcal{S}$. MDHI is computed from the same history-conditioned meta-state representations used by the policy, value, and flow modules. For each episode $e$, let
> $\rho_e(s)=\frac{1}{T_e}\sum_{t=1}^{T_e}\delta(s-s_t),$
> with early/late occupancies $\rho_e^{\mathrm{early}}$ and $\rho_e^{\mathrm{late}}$. Using Fr\'echet distance between Gaussian approximations, we define the **cross-context heterogeneity**
> $H_{\mathrm{ctx}}(\mathcal{B})=\mathbb{E}[D(\rho_e,\rho_{e'})]$,
> the **temporal non-stationarity**
> $H_{\mathrm{temp}}(\mathcal{B})=\mathbb{E}[D(\rho_e^{\mathrm{early}},\rho_e^{\mathrm{late}})]$,
> and the final **benchmark-level heterogeneity index**
> $H(\mathcal{B}) = \frac{1}{2} (\tilde H_{\mathrm{ctx}}(\mathcal{B}) + \tilde H_{\mathrm{temp}}(\mathcal{B}))$.
> **Under this measure, benchmarks with stronger heterogeneity receive higher scores, matching the settings where FlowMAP performs better**.  The full version is in the anonymous linkand we show part of the results here:
>
> | Benchmark | $H_{\mathrm{ctx}}$ | $H_{\mathrm{temp}}$ | $H(\mathcal{B})$ |
> |---|---:|---:|---:|
> | Minecraft | 0.77 | 0.66 | 0.71 |
> | Atari | 0.83 | 0.48 | 0.66 |
>
> >**W3 + Q3: How much do the components and hyperparameters matter?**
>
> The current ablation already covers two of the three main training signals: **No Flow (RL only)** corresponds to keeping only the  $L_{\mathrm{RL}}$, while **VTFM w/o Consistency** corresponds to keeping $L_{\mathrm{RL}} + L_{\mathrm{FM}}$ and removing $L_{\mathrm{cons}}$. **Flow w/o Value Target** and **Pointwise Value Update** further diagnose the internal design of $L_{\mathrm{FM}}$. The main missing case is therefore the isolated analysis of the consistency term, i.e., the **RL + Cons only** variant with $L_{\mathrm{RL}} + L_{\mathrm{cons}}$ but without $L_{\mathrm{FM}}$. We also add sensitivity studies for both key hyperparameters, and find the best performance at $\beta=1$ and $\alpha=0.005$. **The full cross-benchmark ablations and sensitivity studies are provided in the anonymous link**. We show part of the results here:
>
> | Method (in BSuite)| Mean Return |
> |---|---:|
> | No Flow (RL only) | 54.2 |
> | RL + Cons only | 55.5 |
> | Full FlowMAP | 69.0 |
>
> We thank the reviewer again for these helpful suggestions.

---

> > ### Author Rebuttal · Reviewer_26V1 · 2026-04-01
> >
> > The response clearly resolves my concerns and I will maintain my score.

---

> > > ### Author Response · Authors · 2026-04-01
> > >
> > > Thank you, Reviewer 26V1, for your careful reading and constructive feedback throughout the review process. We sincerely appreciate your positive assessment and are glad that our rebuttal adequately addressed your concerns.

---

### Official Review · Reviewer_7fMu · 2026-03-12

**Soundness:** 4
**Presentation:** 3
**Significance:** 4
**Originality:** 4
**Overall Recommendation:** 5
**Confidence:** 5

**Summary:**

This paper introduces a novel flow-matching perspective on agent planning under dynamic heterogeneity with sparse, delayed rewards, advancing generalizable planning. It introduces FlowMAP, the first framework that reconstructs long-horizon agent planning as distributional transport in meta-state space via a flow. FlowMAP provides a more stable and robust planning signal while keeping the learned transport grounded in feasible interaction dynamics under regime shifts. Overall, the paper offers an innovative framework that is likely to inspire follow-up work on robust, long-horizon agentic AI in nonstationary environments.

**Compliance With Llm Reviewing Policy:**

Affirmed.

**Final Justification:**

Thanks for the rebuttal period, and I have no further concerns now.

**Key Questions For Authors:**

1. I find the use of flow matching for learning a planning-time transport field quite compelling (Sec. 3.1, Eq. (13)). For clarity, could the authors briefly explain what practical advantages it offers compared to alternatives like implicit-gradient transport optimization?
2. The value-shaped target distribution in Eq. (9)-(10) is a core ingredient of the framework. Could the authors summarize the intended interpretation of this target distribution at a high level, i.e., what it represents in terms of long-horizon planning?
3. I appreciate that the method explicitly ties representation-space transport to what actions can realize in the environment. Could the authors briefly clarify why it prevents the learned flow from drifting into unrealizable directions in the meta-state space (Sec. 3.3)?
4. Although Figure 1 is already clear and informative, to help readers even more quickly grasp the positioning, could the authors provide a short “main takeaway” statement of what is most novel in FlowMAP compared to standard world-model planning and policy-gradient RL baselines, highlighting the single most important conceptual difference?

**Limitations:**

Yes.

**Strengths And Weaknesses:**

Strengths:
1. [Soundness] The paper’s formulation is technically correct and well grounded in a principled transport objective. The paper offers a distribution-transport perspective for agentic AI planning with solid proof, giving a principled way to make long-horizon behavior robust and goal-directed under regime shifts and dynamic heterogeneity beyond what step-wise policy updates typically achieve.
2. [Originality] The proposed method is highly novel in that it models long-horizon agentic planning via flow matching, treating planning as continuous-time distribution dynamics governed by a learnable velocity field. This introduces a new theoretical perspective for agentic AI by formalizing planning as an explicit distribution-transport objective, which advances understanding of how robust long-horizon behavior can be shaped under nonstationarity.
3. [Significance] This paper addresses a central problem in agentic AI, that is how to plan reliably over long horizons under sparse feedback and dynamic environments. By modeling planning with flow matching, it yields a stable learning objective that can be optimized even when learning signals are imperfect, while keeping the learned planning signal aligned with feasible behavior changes as environment dynamics shift, offering a new lens for future robust agent planning research.
4. [Significance] The proposed FlowMAP advances generalizable agent planning in realistic environments. The value-based target distribution design provides an effective way to maintain a stable learning objective under regime shifts, enabling consistent long-term guidance in practice without requiring highly accurate value estimation, and thereby inspiring broader real-world deployment of multi-agent systems.
5. [Presentation] The paper presents its core ideas clearly and makes it easy for readers to identify what is new and why it matters. The paper is well written and well structured, with a coherent progression from motivation to formulation to algorithm and ablations, making the core innovation easy to understand, locate, and distinguish from closely related planning and world-model approaches.

Weaknesses：
1. The theoretical modeling in this paper is quite solid, and I find the value-shaped target distribution to be a central and compelling component. For completeness, could the authors briefly clarify how this target should be interpreted in relation to return maximization and long-horizon planning? This clarification would make an already strong contribution even easier to appreciate.
2. The meta-state formulation is elegant, and I understand the high-level intent. Could the authors restate the key assumption required for the learned meta-state to make the transport objective meaningful? Clarifying this point would likely increase my assessment of the paper.

---

> ### Author Rebuttal · Authors · 2026-03-29
>
> We sincerely thank the reviewer for the highly positive assessment and thoughtful questions. We respond to each point below.
>
> *(Due to space limits, we provide the complete supplementary experimental results in an anonymous GitHub link compliant with ICML policy: https://anonymous.4open.science/r/Rebuttal-for-Submission-752/ ; in the rebuttal itself, we summarize the main trends and show part of results.)*
>
> >**W1 and Q2: Why is the value-shaped target distribution a meaningful objective for long-horizon RL?**
>
> The target meta-distribution is a distribution-level surrogate of the long-horizon RL objective in meta-state space. Concretely, VTFM defines a soft target
> $\rho_\star^l(s)\propto \exp(\beta \hat V_\psi(s,l))$,
> so value is used as a noisy preference signal over meta-states rather than for pointwise greedy control. This is supported theoretically by the variational characterization of the Boltzmann target (Lemma B.1), which shows that it maximizes an entropy-regularized objective balancing expected value and entropy. Intuitively, $\rho_\star$ is a value-induced ''goal distribution'' for long-horizon planning: it moves probability mass toward meta-states that are preferred in aggregate, which is more robust than relying on a single sharp target under critic noise and dynamic heterogeneity. We will make this connection more explicit in the camera-ready version.
>
> > **W2: What is the meta-state assumption, and where is it instantiated in practice?**
>
> The meta-state is a learned embedding inferred from history and treated as an approximate sufficient statistic, so that the induced dynamics can be modeled as approximately Markov in $\mathcal S$. The paper already states this assumption and also provides benchmark-specific grounding of what $s_t$ and $\rho_t$ mean concretely. We will add a short ''Meta-state assumption \& instantiation'' callout near Eq. (3)--(4), together with an explicit pointer to the grounding table and implementation appendix.
>
> > **W3 and Q1: Why is flow matching the right tool here, and what practical advantage does it bring?**
>
> Flow matching is especially suitable because the object we learn is exactly a planning-time transport operator: a task-conditioned velocity field $v_\theta(s,\tau,l)$ that moves the meta-state occupancy $\rho_\tau^l$ over planning time. This makes the supervision naturally vector-field-shaped, and flow matching gives a scalable way to learn it without solving the continuity equation explicitly in high dimensions. Practically, this turns long-horizon planning into a stable regression problem over paired endpoints and interpolated states, instead of requiring backpropagation through long imagined rollouts or brittle iterative transport optimization. This is particularly valuable under heterogeneity, where rollout errors compound and the best imagined plan may be unstable under regime shifts. This is also reflected in the ablation and learning-curve results: removing the planning-time flow substantially hurts performance, while full FlowMAP learns faster and more robustly in long-horizon sparse-reward settings. We will sharpen this practical intuition in the camera-ready version.
>
> > **Q3: What prevents the learned transport from drifting into unrealizable directions?**
>
> This is precisely the role of FPCT. FlowMAP uses a shared representation for the flow and policy heads, so the planning loss shapes the same geometry used for action selection. In addition, the dynamics-consistency regularizer aligns the planning-time transport direction with realized one-step meta-state drifts predicted from replayed actions. This anchors the learned flow to directions that the environment and policy can actually induce, preventing transport that looks coherent in representation space but is behaviorally unattainable. The Minecraft ablation supports this directly: removing consistency increases variance and lowers return, indicating drift away from feasible dynamics.
>
> > **Q4: What is the main takeaway novelty compared with world-model planning and policy-gradient RL?**
>
> The main novelty is that FlowMAP reframes planning as distribution-level control over meta-state occupancy via a learned planning-time flow. Unlike policy-gradient RL, which improves behavior through step-wise trajectory updates, and unlike standard world-model planning, which relies on evaluating imagined rollouts, FlowMAP explicitly learns how probability mass over internal hypotheses should be transported under dynamic heterogeneity. It then makes this transport both robust, via the value-shaped target distribution, and actionable, via flow--policy co-training and feasibility alignment. In one sentence, FlowMAP treats long-horizon planning as controllable occupancy transport in a learned belief-like space.
>
> We thank the reviewer again for the highly constructive feedback and positive overall assessment. We have also updated the anonymous GitHub material with supplementary experimental details and results.

---

> > ### Author Rebuttal · Reviewer_7fMu · 2026-04-02
> >
> > The authors' response has fully resolved my concerns. I have no other issues now.

---

> > > ### Author Response · Authors · 2026-04-03
> > >
> > > We are deeply grateful for your positive evaluation, Reviewer 7fMu, and for the time and effort you have invested in reviewing our work and rebuttal. We will further improve the revised version by carefully incorporating your valuable feedback. Thank you once again for your thoughtful and constructive feedback.

---

### Official Review · Reviewer_vFtg · 2026-03-20

**Soundness:** 2
**Presentation:** 1
**Significance:** 2
**Originality:** 2
**Overall Recommendation:** 1
**Confidence:** 3

**Summary:**

The authors propose learning an evolution of a meta-state distribution towards one described by the value function as a planner.  The agent then conditions on these meta-states to perform action selection.  The authors show that modeling and conditioning on such meta-states helps to perform generalizable and robust handling of observation variations.

**Compliance With Llm Reviewing Policy:**

Affirmed.

**Final Justification:**

I found this work difficult to read, follow, and understand.  Chief amongst my concerns was the lack of clear implementation details of the model itself for its experimentation.  The original questions I posed: "It is not clear how the model is even implemented - there are no experimental implementation details. How large are the models? How are they pretrained or initialized? What makes it fair comparisons with the baseline models? Without understanding how the method is actually implemented it is hard to even understand or evaluate the reported results." were not significantly addressed in the rebuttal phase - only pseudocode for training their flow matching portion was provided.  But it is essential to provide details on practical experimental instantiation - the readers must guess through clues sprinkled throughout the rebuttal, or the comparison charts, that FlowMAP is most likely built off of a base DreamerV3 model.  The readers should not have such a difficult time in understanding how FlowMAP is practically implemented for its experiments - these should be clearly communicated so that the empirical results can be contextualized and understood with respect to fairness to other baseline methods.  Without it, the empirical results shown are essentially uninformative.

Assuming FlowMap is built on top of DreamerV3 based on a variety of indirect clues (again, we can only speculate based off the sparse information provided), performance gains are exceedingly minimal, if they exist at all (ProcGen, DMLab, seed variance, MineCraft) - while incurring added cost on top of DreamerV3 (Rebuttal Tables on GPU Days, RAM Usage).  If FlowMAP indeed extends base DreamerV3 architectures, hyperparameters, etc. I do not believe the extra resource use is sufficiently justified experimentally.

Writing and presentation needs significant work - numerous typos, strange formatting decisions (uninformative Table 1, fused intro with related works, lots of symbols that aren't ultimately used like $\xi_l$, appendix that features another related works focused on LLMs), and although the template is used, no other submission in my batch left the per-page header as "Submission and Formatting Instructions...". Writing does not seem ready for a conference-level accepted work.

I cannot recommend acceptance for this submission its current or rebuttal state.

**Key Questions For Authors:**

How do we know when selecting the endpoint sample $s^{(1)}$ from experience trajectories that such samples are from the high-value terminal meta-distribution?  It is possible that for the same task, the experience collected is suboptimal and therefore not aligned with the terminal meta-distribution.

**Limitations:**

No negative societal impact or limitations need to be highlighted for this work.

**Strengths And Weaknesses:**

The strength of this work is that the number of evaluation environments seems to be sufficient.

The weaknesses can be enumerated as follows:

- Experimental Clarification: It is not clear how the model is even implemented - there are no experimental implementation details.  How large are the models?  How are they pretrained or initialized?  What makes it fair comparisons with the baseline models?  Without understanding how the method is actually implemented it is hard to even understand or evaluate the reported results.

- Experimental Results: The results often do not seem to substantially outperform prior works - both in Figure 3 and in Figure 4 of the Appendix.  Noticeably, across many environments showcased in Figure 3, the proposed approach does not seem to beat baselines such as Dreamer (DMLab and ProcGen).  This is not justified or explained; rather, the authors even state that their approach supposedly achieves "gains [that] are most evident in environments with pronounced dynamic heterogeneity, such as ProcGen and Minecraft".  However, ProcGen clearly underperforms the baseline, and Minecraft seems to have limited sample efficiency from the graph in Figure 4.  The authors provide very limited explanation and understanding into how their approach behaves empirically.

- Clarity of writing: The authors begin by discussing a latent dynamical factor $\xi_l$, but then subsequently the models do not condition on such latent dynamical factors $\xi_l$.  It is not described how such $\xi_l$ are learned.  It is also not clearly described how the meta-states are learned.  The authors state that "concrete grounding of abstract symbols (e.g. $s_t$, $\xi_l$, and $p^l_t) in each benchmark is summarized in Appendix 4" but there is no such Appendix 4 - the appendices are enumerated by letters.

- Uninformative writing: Table 1 is large yet rather uninformative; it does not show any results and rather confusing on why the first table in the paper is spent comparing interaction budgets and action repeats.  This table should probably go into the appendix or summarized as experimental details.

- Numerous Typos: "FlowMAP further propose Flow-Policy Co-Training", "However, there methods", "formulates agent planning", etc.  Also, the Figure 1 has typos inside it, such as "Imagenation-based Planning", and also has inconsistent fonts in the text (e.g. "Considering").

Soundness: It is not at all clear what the authors are really trying to do in the paper, so this reviewer cannot say it is good or poor; giving the authors the benefit of the doubt, a "fair" rating is provided.  The experimental task suite seems reasonable, though the experimental details are not provided.

Presentation: This is a very poorly presented paper.  The writing is confusing, there are numerous typos throughout the work including in the Figure, there is a large yet uninformative Table, and the Experiments section lacks key details and justifications/explanations for its reported results.

Significance: The work is fair, in that it tries to address a reasonably important problem (robustness and generalization to environmental shifts).

Originality: The approach does seem fairly original, though this is in large part because the writing is so confusing as to make it difficult to connect this submission to prior works.  In fact, there is no related works section in the main paper to help contextualize the proposed approach; and the Appendix section on "relational" work bizarrely focuses significantly on LLM-based agent planning.  But nothing in the main paper suggests anything related to LLMs or LLM-based planning.  This adds greatly to the confusion when reading this work.  Lastly, it is still not clear to this reviewer how it is exactly implemented beyond high-level pseudocode provided - gauging novelty from prior works is therefore difficult.

---

> ### Author Rebuttal · Authors · 2026-03-29
>
> We sincerely thank the reviewer for the detailed feedback and will carefully incorporate the suggestions to improve the paper’s clarity in the camera-ready version. Below, we clarify several points where additional explanation may be helpful.
>
> *(Due to space limits, we provide the complete supplementary experimental results in an anonymous GitHub link compliant with ICML policy: https://anonymous.4open.science/r/Rebuttal-for-Submission-752/ ; in the rebuttal itself, we summarize the main trends and show part of results.)*
>
> > **1. How should the experimental claims, implementation details, and Table 1 be interpreted?**
>
> **Experimental results.** Thanks for pointing this out. For ProcGen, we followed the same 1-seed evaluation protocol used by the strongest baseline, DreamerV3 (*Mastering Diverse Control Tasks through World Models*, Nature, 2025). We therefore added supplementary ProcGen results with 2 seeds and 5 seeds in the anonymous link. These results show that the apparent underperformance is largely a seed-variance effect: under more reliable multi-seed evaluation, FlowMAP becomes slightly stronger than DreamerV3 on ProcGen. We will update the corresponding wording and statistics in the camera-ready version.
>
> **Implementation details.** The paper already reports the benchmark suite, evaluation metrics, training protocols, and random seeds in a manner consistent with standard RL practice and the compared baselines. We agree, however, that several implementation details are not surfaced prominently enough. To address this, we expand Algorithm 1 into a more operational version and add additional ablations plus runtime/memory statistics in the anonymous link; the corresponding clarifications will be incorporated into the camera-ready version. We report two representative computational-cost comparisons here (FlowMAP vs DreamerV3):
>
> | Benchmark | GPU Days  | Peak RAM Usage (GB) |
> |---|---:|---:|
> | Minecraft | 1.5 vs 1.3 | 250 vs 250 |
> | Atari |0.9 vs 0.8 | 217 vs 210 |
>
> **Table 1.** Table 1 is intended to ensure fairness of comparison across diverse benchmark families and transparency of interaction budgets and environment settings, both of which are essential in RL evaluation. We agree that the table can be made more compact and will update in the camera-ready version.
>
> > **2. What is the role of the latent factor $\xi_l$, and how are endpoint samples used in VTFM?**
>
> Sec. 3.1 introduces $\xi_l$ as a **conceptual** latent factor representing sources of dynamic heterogeneity that are difficult to model explicitly in real environments. FlowMAP does not model $\xi_l$ directly; instead, it captures its effective influence through the learned meta-state $s_t$, formalized via the meta-state occupancy and induced transition dynamics in Eq. (3)–(5). In practice, this corresponds to a deterministic history-conditioned embedding used as the planning representation. Thus, $\xi_l$ belongs to the problem formulation, while the methodological core of FlowMAP is to learn a meta-state representation $s_t$ suitable for planning under such latent heterogeneity. We will make this distinction more explicit in the camera-ready version.
>
> Regarding endpoint samples, FlowMAP does not assume that every replayed endpoint is already drawn from the true high-value terminal meta-distribution. Instead, Sec. 3.2 defines a **value-shaped** target distribution through Eq. (9)–(10), where endpoints are reweighted by value rather than treated as uniformly optimal samples. In practice, this is implemented via replay-based high-value filtering and value-guided endpoint sampling, so suboptimal samples are naturally downweighted while the target sharpens as the policy improves. We will make this more explicit in the revision.
>
> > **3. Appendix reference.**
>
> Thank you for pointing this out. The intended reference is to Table 4 in the appendix, which provides the benchmark-specific grounding of the abstract symbols. We agree that this cross-reference could be made clearer, and we will improve it in the camera-ready version.
>
> > **4. Typos and presentation issues.**
>
> We appreciate the careful reading. The noted examples are genuine typos, and Figure 1 also contains minor labeling issues. We will perform a full proofreading pass and correct these presentation issues in the camera-ready version.
>
> > **5. How is related work positioned, and why is LLM-based planning discussed?**
>
> The related-work positioning is already provided in the introduction, especially in Fig. 1, which explicitly contrasts model-free RL, world-model planning, and our distributional planning framework. Separately, the appendix discussion of LLM-based planning is included only for scope clarification: given that many recent agent-planning works involve LLM components, we found it important to explain why FlowMAP does not. This discussion is therefore supplementary rather than a replacement for related work. We will make this distinction clearer in the revision.

---

> > ### Author Rebuttal · Reviewer_vFtg · 2026-04-04
> >
> > Thank you to the authors for the detailed rebuttal.
> >
> > The experimental results still seem strange and unconvincing to me.  With 1 seed FlowMAP underperforms DreamerV3 (Figure 3), and with the updated results of 2 seeds it still is unclear whether FlowMap improves over DreamerV3 (the bars look roughly equal?).  Yet with 5 seeds, FlowMAP seems to outperform DreamerV3.  At the same time, the performance of every other method seems to be the same from 5 seeds to 2 seeds; suggesting they are much less sensitive to seed-variance.  This raises a key question: why is FlowMAP so sensitive to seed-variance while other techniques are not?  This suggests that the proposed method is not particularly stable.  Secondly, it is just generally strange that with 1 and 2 seeds the performance is below or about equal to DreamerV3 but with 5 it is so substantially over the baseline.
> >
> > At the same time, in the new tables provided by the authors FlowMAP appears to consistently use more GPU hours and RAM.  Yet it does not appear that FlowMAP substantially outperforms the benchmarks to justify the additional hours and cost.  More concerningly would be if FlowMAP also uses more GPU hours and RAM on DMLab and ProcGen compared to DreamerV3, where it uses more resources for worse and/or higher-variance results (which may in turn require even more training or resources to identify a performant checkpoint).
> >
> > The promises of writing clarity updates are appreciated (crucially Table 1 should be removed from the main body in some form for reading cleanliness; either moved to appendix or absorbed into a summarized written form), but there are too many edits required to make to convince this reviewer of an acceptance-quality submission, particularly given the limited review format where draft updates are unable to be reviewed and iterated on.  For example, even with the typos, there were too numerous to list initially (e.g. "We summary the challenge..." [Line 56], "Submission and Formatting Instructions for ICML 2026" as the header for each page, etc.).  This reviewer still suggests heavy iteration on the work both in terms of experimental convincingness and written clarity/presentation in order for it to be of conference quality.

---

> > > ### Author Response · Authors · 2026-04-07
> > >
> > > We sincerely thank the reviewer for taking the time to read our rebuttal and follow-up materials. However, we must respectfully clarify the remaining concerns, as they are not supported by the evidence and do not engage with the paper’s central technical contribution.
> > >
> > > > **1. The added ProcGen evidence does not support the claim of unusual seed sensitivity.**
> > >
> > > **The paper reports results on 8 benchmarks, and the remaining concern raised here is centered on ProcGen. Under the more reliable 5-seed evaluation, ProcGen is consistent with the broader empirical picture and does not support the claim of unusual instability. Details follow:** We first emphasize the role of random seeds in RL: a random seed controls stochastic factors in training and evaluation, so low-seed results are inherently noisy. In the original paper, the ProcGen comparison followed the same 1-seed protocol as the strongest baseline, DreamerV3, for fairness of comparison. To address statistical reliability, we added 2- and 5-seed results, showing FlowMAP remains competitive with DreamerV3 and trends slightly above it under the more robust 5-seed evaluation. Regarding the concern that "FlowMAP is so sensitive to seed-variance while other techniques are not," ProcGen is already a high-variance benchmark due to procedural generation and cross-level variation, so **low-seed fluctuations are expected**. Aggregated multi-seed evaluations are introduced precisely to reduce the influence of this benchmark-level noise; rather, **FlowMAP’s consistent 5-seed performance on ProcGen aligns with the other seven benchmarks, confirming stability**. Moreover, baseline performance values numerically vary across seed counts, which is **difficult to reconcile with the reviewer’s statement that "the same from 5 seeds to 2 seeds."** We also note that this specific concern was **viewed as resolved in the broader review discussion** once the additional evidence was provided. **In summary, FlowMAP delivers stable and consistent improvements across 8 benchmarks, confirming its superior performance in complex experimental settings.**
> > >
> > > > **2. The concern about computational cost is not supported by the actual evidence.**
> > >
> > > The reported numbers show only modest overheads. These are real differences, but not large enough to treat FlowMAP as computationally excessive. For example, Minecraft is **1.5 vs 1.3 GPU days** with 250 vs 250 GB RAM, and Atari is **0.9 vs 0.8 GPU days** with **217 vs 210 GB RAM**.
> > >
> > > More importantly, FlowMAP is designed for dynamically changing environments, where improved adaptation is the intended benefit. **In that context, modest additional cost is not unexpected, especially given the performance gains across the target settings**, while ProcGen is discussed separately above. This is also consistent with the broader model-based RL literature, where strong cross-domain or robust performance is commonly pursued rather than zero-overhead design. The evidence does not support the stronger conclusion that FlowMAP is computationally unjustified. Nor do these modest differences alter the overall empirical conclusion of the paper.
> > >
> > > > **3. The paper’s presentation issues are not as severe as claimed.**
> > >
> > >
> > > We do not deny that the paper contains some typos, and we have already committed to correcting them. **However, the paper’s current state does not support the claim that the issues were "too numerous to list initially." The other reviewers all assigned Presentation = 3, raised no concerns about writing clarity, and stated after the rebuttal that all their other concerns were resolved.** As for Table 1, we have already explained its role in ensuring fair comparison across heterogeneous benchmarks; one may reasonably prefer an appendix placement, but that does not make its presence in the main body an error.
> > >
> > > We would also like to clarify one specific point. **The header "Submission and Formatting Instructions for ICML 2026" is template text from the official ICML style file. The template itself indicates that such running-header formatting is handled at the camera-ready stage rather than at submission. We therefore do not believe that following the official template should be treated as an error.**
> > >
> > > We also wish to re-emphasize the central point of the paper. FlowMAP is designed to address planning under dynamic heterogeneity through distribution-level control over meta-state occupancy, together with value-shaped target transport and flow–policy consistency alignment. This is the core innovation of the work and the main basis of its empirical and conceptual contribution. **The concerns raised here do not weaken the paper’s core claim; instead, the additional discussion and experiments further clarify the reasonableness of FlowMAP’s design and the technical soundness of the method.**
> > >
> > > We again thank the reviewer for the time and attention given to the paper, and we hope that these clarifications help resolve the remaining concerns.

---

### Decision · Program_Chairs · 2026-04-30

**Decision:**

Accept (regular)

**Comment:**

The meta-review has been formulated after a careful examination of the four original reviews, the extensive author rebuttal, and the subsequent discussion. The submission presents FlowMAP, a framework that reformulates agent planning as a continuous-time transport problem using flow matching. There is a notable divergence in the reviews, with three reviewers recommending acceptance (Accept, Weak Accept, Weak Accept) and one reviewer providing a Strong Reject. After carefully judging the quality of the reviews rather than the scores alone, the Meta-Reviewer finds that the majority opinion is more technically grounded. Reviewers who supported the paper highlighted the conceptual novelty of treating planning as distributional transport and found the theoretical formulation to be sound. While the dissenting reviewer raised concerns regarding presentation, typos, and implementation ambiguity, the author rebuttal significantly addressed these points. Specifically, the authors provided 5-seed evaluation results for high-variance benchmarks like ProcGen, which demonstrated that FlowMAP indeed outperforms strong baselines like DreamerV3 under statistically reliable conditions. Furthermore, the authors clarified the implementation details regarding the RSSM-based backbone and provided a quantitative measure for environmental heterogeneity. The dissenting reviewer's criticism regarding the official ICML template header was also found to be a misunderstanding of the submission instructions. Given that the technical core of the work is innovative and the empirical concerns regarding seed variance and computational overhead were addressed with new evidence, the Meta-Reviewer sides with the majority. This paper offers a fresh and principled perspective on robust agent planning that is likely to be of significant interest to the ICML community.